# Fast Data Mixture Optimization via Gradient Descent

**Haoru Tan**[1,2], **Sitong Wu**[3], **Yanfeng Chen**[2] ✉, **Jun Xia**[2],
**Ruobing Xie**[2], **Bin Xia**[3], **Xingwu Sun**[2], **Xiaojuan Qi**[1] ✉

[1]The University of Hong Kong   [2]Hunyuan Team, Tencent   [3]The Chinese University of Hong Kong

## Abstract

While large and diverse datasets have driven recent advances in large models, identifying the optimal data mixture for pre-training and post-training remains a significant open problem. We address this challenge with **FastMix**, a novel framework that automates data mixture discovery while training only a *single proxy model*. Instead of relying on predefined heuristics or resource-intensive simulations, FastMix jointly optimizes mixture coefficients and model parameters, substantially improving efficiency and scalability over prior approaches. At the core of FastMix is a reformulation of mixture selection as a *bilevel optimization* problem. Under this reformulation, we show that optimizing mixture ratios is mathematically equivalent to assigning per-source loss weights under uniform source sampling. This embeds the mixture coefficients directly into the differentiable iterative optimization objective, enabling efficient, gradient-based optimization of both mixture and model. To solve the optimization problem, FastMix implements an approximate iterative optimization procedure, alternating between (i) updating model parameters on data sampled according to current mixture ratios (inner loop) and (ii) updating mixture ratios based on validation feedback (outer loop). Across pre- and post-training, FastMix outperforms baselines while drastically reducing search cost: in pre-training, it attains an average score of **48.2** with **1.3** GPU-hours (×**550** vs. RegMix; ×**55** vs. CLIMB), and in post-training (SFT) it leads with **65.4** with a +5.5 gain over the next best, completing search in **2.2** GPU-hours compared to the 115 GPU-hours required by CLIMB/RegMix.

## 1 Introduction

The performance of large-scale models (Yang et al., 2024b; Dubey et al., 2024; Touvron et al., 2023; Hu et al., 2024) depends critically on the data used for training. While large and diverse datasets have driven recent advances, identifying the optimal data mixture for pre-training (Shukor et al., 2025) and post-training (Dong et al., 2023) remains a significant challenge.

Popular methods such as manual trial-and-error (Yang et al., 2023; Tong et al., 2024) or proxy-based methods (Liu et al., 2024; Diao et al., 2025) often do not scale well as models grow larger. For example, proxy-based search methods such as RegMix (Liu et al., 2024) and CLIMB (Diao et al., 2025) have demonstrated strong generalization and stability, yet they require training a large number of proxy models during the search. This results in prohibitive computational overhead, making mixture optimization increasingly impractical as both models and datasets continue to expand. The central question is thus: how can we efficiently determine effective data mixtures for large-scale training?

We address this challenge with FastMix, a novel framework that automates data mixture discovery while training only a *single proxy model*. Instead of relying on predefined heuristics or resource-intensive simulations, FastMix jointly optimizes mixture coefficients and model parameters, substantially improving efficiency and scalability over prior approaches. At the core of FastMix is a reformulation of mixture selection as a weighted *bilevel optimization* problem in Eq.(2). Specifically, we show that optimizing mixture ratios is mathematically equivalent to assigning per-source loss weights under uniform source sampling. This reparameterization embeds the mixture coefficients directly into the differentiable iterative optimization objective, enabling efficient, gradient-based optimization of both mixture and model. To solve the optimization problem (Maclaurin et al., 2015;

---

✉ Corresponding author: Yanfeng Chen and Xiaojuan Qi

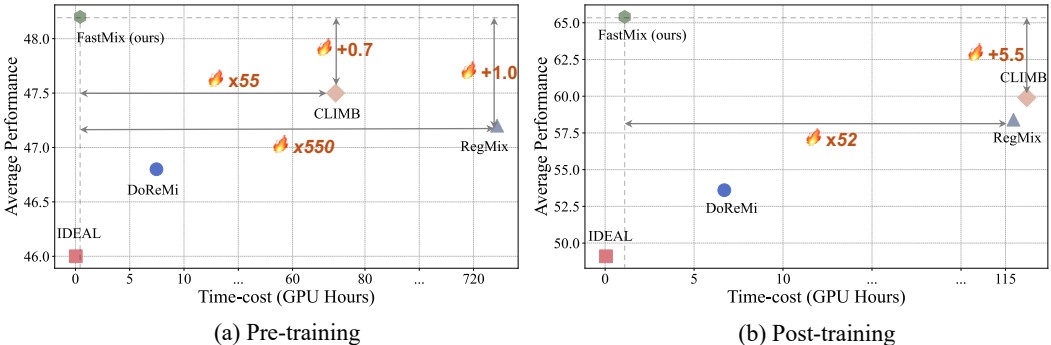

(a) Pre-training  (b) Post-training

Figure 1: Average Performance versus Time-cost (GPU Hours) comparison for various data mixture strategies. **(a) Pre-training:** Our proposed FASTMIX *(ours)* method achieves the highest performance with the lowest time-cost. The annotations highlight that it is up to *55×* more time-efficient than *CLIMB* (Diao et al., 2025) and **550×** more time-efficient than *RegMix* (Liu et al., 2024), while providing a significant performance gain. **(b) Post-training:** In this setting, FASTMIX *(ours)* again demonstrates state-of-the-art performance and time-efficiency, outperforming *RegMix* with a **52×** reduction in time-cost and gaining an additional **5.5** performance points over *CLIMB*. This illustrates the superior trade-off between performance and time cost achieved by our method.

Franceschi et al., 2018), FASTMIX implements an approximate iterative optimization procedure, alternating between (i) updating model parameters on data sampled according to current mixture ratios (inner loop) and (ii) updating mixture ratios based on validation feedback (outer loop) via a gradient-based optimizer (Kingma & Ba, 2014).

Extensive evaluations demonstrate that FASTMIX optimizes data mixtures across model scales and tasks in both pre-training and post-training, outperforming baselines at a fraction of the computational cost (See Fig. 1). In pre-training, it delivers a top average score of 48.2 and rank 1 across 14 benchmarks (best on 9) with just **1.3** GPU-hours, achieving ×**550** faster than RegMix (Liu et al., 2024) and ×**55** than CLIMB (Diao et al., 2025). In post-training (SFT), a math-tuned mixture generalizes to coding and STEM-QA, reaching 65.4 (**+5.5** over next best) in **2.2** GPU-hours versus more than **115** GPU-hours for CLIMB/RegMix. Overall, FASTMIX makes mixture optimization practical and scalable for next-generation large models.

## 2 RELATED WORK

Data-centric technologies Tan et al. (2025a); Xia et al. (2025); Tan et al. (2023); Wu et al. (2024); Tan et al. (2025b); Wu et al. (2025a); Tan et al. (2025c) play a crucial role in the recent rapid progress of large models (Dubey et al., 2024; Touvron et al., 2023; Allal et al., 2024; Yang et al., 2023; 2024a; Zhang et al., 2025b; 2026; 2025a; Wu et al., 2025b). The training of modern large models relies heavily on strategically mixing data from diverse sources, spanning languages (Yang et al., 2023), modalities (Gunasekar et al., 2023; Yang et al., 2024b), and difficulty levels (He et al., 2025). This *data mixture problem* (Ge et al., 2024) presents fundamental challenges not only in pre-training (Shukor et al., 2025; Dubey et al., 2024; Yang et al., 2024b) but also in post-training (Dong et al., 2023; Ming et al., 2025; Tong et al., 2024). Early practice largely relied on manual heuristics, which lack standardization and often fail to generalize across settings. More recently, optimization-based approaches (Xie et al., 2024; Fan et al., 2023; Liu et al., 2024) have been introduced to automate mixture selection.

**Proxy-based methods** (Xie et al., 2024; Liu et al., 2024; Diao et al., 2025) adopt a two-phase design in which a proxy model is trained under candidate mixtures and its performance is used to infer optimal sampling ratios. For example, DoReMi (Xie et al., 2024) trains a small proxy to adjust domain weights based on relative losses, then reuses the optimized ratios to train a larger model. RegMix (Liu et al., 2024) scales this idea by training hundreds of proxy models under different ratios, fitting a regression model on the resulting mixture-performance pairs, and extrapolating the optimal mixture. CLIMB (Diao et al., 2025) improves efficiency by iteratively refining the search region, reducing the number of proxy models required. Other works (Ye et al., 2024; Shukor et al., 2025;

Kang et al., 2024) study cross-scale transfer: Shukor et al. (2025) provide theoretical and empirical evidence that mixtures found on small models generalize to larger ones, while Ye et al. (2024); Kang et al. (2024) report functional relationships between mixture proportions and performance.

In contrast, **dynamic methods** (Chen et al., 2024; Ming et al., 2025; Albalak et al., 2023) remove the separate search phase by adjusting mixtures on the fly. IDEAL (Ming et al., 2025), for instance, leverages influence functions (Koh & Liang, 2017) to estimate domain contributions to downstream performance and to dynamically rebalance training data.

Overall, proxy-based methods such as RegMix and CLIMB generally achieve stronger and more stable performance than dynamic approaches, but at substantial computational cost. Our method, FASTMIX, preserves the reliability of proxy-based optimization while cutting search time from hundreds of GPU-hours to nearly one, achieving both higher efficiency and stronger generalization.

## 3 FASTMIX

### 3.1 PROBLEM REFORMULATION WITH REPARAMETERIZATION

**Data Mixture as a Bi-level Optimization Problem.** Formally, data mixture optimization can be posed as a bilevel optimization problem. Let $D = \{D_1, \ldots, D_k\}$ be a collection of data sources (or clusters), and let $\alpha \in A \subset \mathbb{R}^k$ denote the mixture weights, where the feasible set $A$ is the probability simplex ($\alpha_i \geq 0$ and $\sum_{i=1}^{k} \alpha_i = 1$). Given mixture $\alpha$ and model parameters $w$, the training objective is $\mathcal{L}_{\text{train}}(D, w \mid \alpha)$. Let $w^*(\alpha)$ be the parameters obtained by (approximately) optimizing this training objective under $\alpha$. The target is to find mixture weights $\alpha^*$ that minimize the validation loss, i.e., $\mathcal{L}_{\text{target}}(w) = \ell_{\text{val}}(V, w)$ evaluated at $w^*(\alpha)$:

$$\min_{\alpha} \ \mathcal{L}_{\text{target}}\Big(w^*(\alpha)\Big) \quad \text{s.t.} \quad w^*(\alpha) = \arg\min_{w} \mathcal{L}_{\text{train}}\Big(D, w|\alpha\Big), \ \sum_{i=1}^{k} \alpha_i = 1, \ \alpha_i \geq 0. \quad (1)$$

where the inner-loop aims to find the optimal model weights $w^*(\alpha)$ by minimizing the training loss on the dataset given mixture weights $\alpha$. The outer-loop then seeks to optimize these mixture weights $\alpha$ to minimize the model's final loss on target tasks.

While the bi-level formulation is conceptually appealing, it is difficult to solve in practice. The crux is handling the mixture weights $\alpha$. Unlike model parameters $w$, which admit efficient gradient-based updates, mixture (sampling) ratios are typically non-differentiable, precluding end-to-end backpropagation. Consequently, practitioners resort to greedy heuristics or policy-gradient (score-function) updates to adjust $\alpha$. These procedures are sample-inefficient and scale poorly with the number of data sources, turning mixture search into a dominant computational bottleneck.

**Differentiable Formulation.** Through a simple reparameterization, we recast the original bilevel problem into a mathematically equivalent, fully differentiable objective. The key idea is to replace stochastic sampling by mixture ratios with per-source, differentiable loss weights applied under uniform sampling, so that each source's contribution is controlled continuously via its weight, yielding the following formulation:

$$\min_{\alpha} \ \mathcal{L}_{\text{target}}\Big(w^*(\alpha)\Big) \quad \text{s.t.} \ w^*(\alpha) = \arg\min_{w} \sum_{i=1}^{k} \alpha_i \mathcal{L}_{\text{train}}\Big(D_i, w\Big), \ \sum_{i=1}^{k} \alpha_i = 1, \ \alpha_i \geq 0, \quad (2)$$

where $\mathcal{L}_{\text{train}}(D_i, w)$ denotes the model's training loss on source $D_i$, computed under *uniform source sampling* (each source selected with probability $1/k$). The inner-loop finds the optimal model weights, $w^*(\alpha)$, by minimizing a weighted sum of the training losses from $k$ different data domains. The data mixture weight $\alpha_i$ serves as the weight for each domain's loss. The outer-loop then aims to optimize these proportions $\alpha$ to minimize the model's loss on target tasks. This reparameterization is key: rather than treating mixture ratios as non-differentiable sampling probabilities, we reinterpret them as continuous coefficients that scale each source's loss. Consequently, the mixture weights $\boldsymbol{\alpha} = (\alpha_1, \ldots, \alpha_k)$ are fully differentiable and amenable to gradient-based optimization. Standard optimizers (e.g., SGD or Adam) can then jointly update the model parameters and the data weights, enabling efficient end-to-end training.

---

**Algorithm 1** FASTMIX Optimization Algorithm

---

1: **Initialize** model parameters $w^0$, mixture weights $\alpha^0$, inner-loop duration $n_1$ and outer-loop duration $n_2$.
2: **for** $t = 0, 1, \ldots, T - 1$ **do**
3:     **if** $(t) \bmod n_1 \neq 0$ **then**
4:         // Inner loop: update model parameters (*e.g.*, via the SGD optimizer, and we can change this update rule to other optimizers, like Adam (Kingma & Ba, 2014))
5:         $w^{t+1} \leftarrow w^t - \eta_w^t \frac{\partial [\sum_{i=1}^k \alpha_i^t \mathcal{L}_{\text{train}}(D_i, w^t)]}{\partial w^t}$,
6:     **else**
7:         // Outer loop: update mixture weights (*e.g.*, via the SGD optimizer, and we can change this update rule to other optimizers, like Adam (Kingma & Ba, 2014))
8:         $\alpha^{t+1} \leftarrow \alpha^t - \eta_\alpha^t \frac{\partial \mathcal{L}_{\text{target}}\left(w^{t+n_2}\right)}{\partial \alpha^t}$
9:     **end if**
10: **end for**
11: **Output**: the optimized mixture weight $a^{\text{final}}$ after the final outer loop update.

---

*Proof of equivalence.* Let $D = \bigcup_{i=1}^k D_i$ denote the union of $k$ data sources (or clusters), and let $\alpha = (\alpha_1, \ldots, \alpha_k)$ be mixture weights with $\sum_i \alpha_1 = 1$, $\alpha_i \geq 0$. To sample a training example $x$, first draw a source index $i \sim \text{Cat}(\alpha)$, then sample $x \sim D_i$. The training loss under this mixture sampling is

$$\mathcal{L}_{\text{train}}(D, w \mid \alpha) = \mathbb{E}_{i \sim \text{Cat}(\alpha)} \mathbb{E}_{x \sim D_i} \big[\ell(x, w)\big] = \sum_{i=1}^k \alpha_i \, \mathcal{L}_{\text{train}}(D_i, w), \tag{3}$$

where $\ell(x, w)$ is the per-example loss and $\mathcal{L}_{\text{train}}(D_i, w) = \mathbb{E}_{x \sim D_i}[\ell(x, w)]$ is the expected loss on source $D_i$. Thus, under mixture sampling, the expected training loss is a convex combination of the per-source losses, with coefficients given by the mixture ratios.

### 3.2 How to obtain better generalization performance?

Like most AutoML algorithms, FASTMIX requires a search target, typically defined as a performance metric on a held-out validation set. However, relying on validation performance alone can lead to overfitting to quirks of the validation data and limited transferability to new scenarios. To improve generalization, we propose two complementary strategies: (i) entropy-based regularization to encourage diversity among mixture weights, and (ii) incorporating training loss into the search target to balance validation and training signals.

**Entropy-based regularization.** Entropy regularization prevents the mixture distribution from collapsing onto a narrow subset of data sources. Given mixture weights $(\alpha_1, \ldots, \alpha_k)$ across $k$ sources, we add the penalty $\mathcal{R}_{\text{entropy}} = \sum_{i=1}^k \alpha_i \log \alpha_i$. Minimizing this term discourages overly peaked distributions, promoting more uniform weight allocation. This reduces sensitivity to spurious validation patterns and improves robustness by leveraging multiple data sources.

**Training loss as an auxiliary target.** We further integrate the training loss into the search objective to complement the validation signal. While the validation term reflects out-of-sample generalization, the training term measures how effectively the model fits the mixture as a whole. Combining the two reduces over-reliance on the limited validation set and guides the search toward mixture ratios that generalize more reliably across both in-domain and out-of-domain data.

**Joint objective.** Together, entropy regularization and the auxiliary training loss yield the following search objective:

$$\mathcal{L}_{\text{target}}(w) = \ell_{\text{val}}(w) + \beta \, \mathcal{L}_{\text{train}}(w) + \lambda \sum_{i=1}^k \alpha_i \log \alpha_i, \tag{4}$$

where $\beta \geq 0$ and $\lambda \geq 0$ are trade-off hyperparameters. Empirically, $\lambda$ is set to a small value (e.g., $10^{-5}$) to encourage diversity without dominating the optimization, while $\beta$ is most effective at moderate values (e.g., $0.1$). We provide a detailed sensitivity analysis of these hyperparameters in our ablation studies. Overall, these two strategies substantially improve the generalization ability of

FASTMIX, enabling it to discover mixtures that not only perform strongly on validation benchmarks but also transfer robustly to broader real-world applications.

## 3.3 OPTIMIZATION

Although the reparameterized formulation enables end-to-end differentiation over both model parameters and data mixtures, the resulting bilevel problem is still difficult to solve directly. Accordingly, we adopt an iterative procedure (Alg. 1) that alternates between updating the model parameters and the mixture weights (Maclaurin et al., 2015; Liu et al., 2018; Pedregosa, 2016; Franceschi et al., 2018). The two key steps are outlined below.

**(i) Inner loop (network parameter update).** Given current mixture weights $\alpha^t$, the model parameters $w$ are updated for $n_1$ steps via stochastic gradient descent (SGD) to minimize the weighted training loss $\mathcal{L}_{\text{train}}$ :

$$w^{t+1} \leftarrow w^t - \eta_w^t \frac{\partial\left(\sum_{i=1}^k \alpha_i^t \mathcal{L}_{\text{train}}(D_i, w^t)\right)}{\partial w^t}, \tag{5}$$

where $\mathcal{L}_{\text{train}}(D_i, w)$ denotes the model's training loss on source $D_i$, computed under *uniform source sampling* (each source selected with probability $1/k$). This is repeated for $n_1$ iterations. Other gradient-based optimizers, such as Adam (Kingma & Ba, 2014), are compatible with our framework. After $n_1$ updates, we denote the resulting parameters as $w^{t+n_1}$.

**(ii) Outer loop (mixture weight update).** The mixture weights $\alpha^t$ are then updated using validation feedback $\mathcal{L}_{\text{target}}$. Specifically, the model is trained for $n_2$ iterations with the previous mixture weights $\alpha^t$, and the resulting parameters $w^{t+n_2}$ are evaluated on the validation loss $\mathcal{L}_{\text{target}}$. The mixture weights are updated as:

$$\alpha^{t+1} \leftarrow \alpha^t - \eta_\alpha^t \frac{\partial \mathcal{L}_{\text{target}}\left(w^{t+n_2}\right)}{\partial \alpha^t}, \tag{6}$$

In effect, $\alpha^{t+1}$ is updated according to how the validation loss responds after $n_2$ steps of training under $\alpha^t$. This naturally assigns larger weights to data sources that contribute more to improving validation performance. A key consideration is how the gradient is estimated, since this directly impacts both the direction of updates and the efficiency of the search.

In the special case $n_2 = 1$ with SGD updates, the gradient of the validation loss with respect to $\alpha_i^t$ yields a closed-form solution:

$$\frac{\partial \mathcal{L}_{\text{target}}\left(w^{t+1}\right)}{\partial \alpha^t} = \frac{\partial \mathcal{L}_{\text{target}}(w^{t+1})}{\partial w^{t+1}} \cdot \frac{\partial w^{t+1}}{\partial \alpha_i^t} = -\eta_w^t \nabla_w \ell_{\text{val}}(V, w^{t+1}) \cdot \nabla_w \mathcal{L}_{\text{train}}(D_i, w^t), \tag{7}$$

where $D_i$ denotes the $i$-th training source. This shows that per-source training losses directly shape the mixture gradients. The following derivation shows why the formula holds. Under the SGD update rule, the weights $w$ at time $t + 1$ are updated based on the gradient of the loss function with respect to the mixture coefficients $\alpha_i^t$: $w^{t+1} = w^t - \eta_w^t \nabla_w [\sum_{i=1}^k \alpha_i^t \mathcal{L}_{\text{train}}(D_i, w^t)]$. Taking the derivative of $w^{t+1}$ with respect to $\alpha_i^t$, we get: $\frac{\partial w^{t+1}}{\partial \alpha_i^t} = \frac{\partial}{\partial \alpha_i^t}\left[w^t - \eta_w^t \nabla_w\left(\sum_{j=1}^k \alpha_j^t \mathcal{L}_{\text{train}}(D_j, w^t)\right)\right]$. Since $w^t$ is independent of $\alpha_i^t$, the derivative of the first term is zero. Due to the linearity of the derivative and the sum, only the term corresponding to $\alpha_i^t$ remains, hence, $\frac{\partial w^{t+1}}{\partial \alpha_i^t} = -\eta_w^t \nabla_w \mathcal{L}_{\text{train}}(D_i, w^t)$.

The formulation in Eq.(7) can be intuitively understood as follows: The gradient with respect to $\alpha_i$ is proportional to the *alignment* between (i) the validation gradient $\nabla_w \ell_{\text{val}}(V, w^{t+1})$ and (ii) the training gradient from source $D_i$, $\nabla_w \mathcal{L}_{\text{train}}(D_i, w^t)$. If these gradients are aligned (positive dot product), the derivative $-\eta_w^t \nabla_w \ell_{\text{val}} \cdot \nabla_w \mathcal{L}_{\text{train}}(D_i, w^t)$ is negative, so a gradient-descent step on $\alpha_i$ *increases* its weight, emphasizing sources whose updates also reduce the validation loss. If they are opposed (negative dot product), the derivative is positive and a step *decreases* $\alpha_i$, down-weighting sources that harm validation performance. Near-orthogonality yields small updates. Thus, the procedure reallocates mass toward data sources whose training signals most effectively improve the validation objective.

When $n_2 > 1$, deriving a closed-form gradient becomes intractable, requiring finite-difference approximations or similar techniques, which are often unstable and inefficient. In contrast, $n_2 = 1$

admits a closed-form gradient that is both computationally efficient and empirically effective. Additional discussion of the multi-step case is provided in the Sec.4.3.1.

### 3.4 Handling Non-Differentiable Cases

Our optimization algorithm is designed for settings where both $\mathcal{L}_{\text{target}}$ and $\mathcal{L}_{\text{train}}$ are differentiable. However, in practice, non-differentiable situations may arise. We discuss two representative cases below.

**Non-differentiable targets.** One common challenge arises when the objective function is non-differentiable, such as when validation performance is measured by discrete metrics (e.g., accuracy) rather than a smooth loss. A standard remedy is to apply finite-difference methods to approximate gradients. However, our experiments show that these methods are often slow and suffer from numerical instability. In such cases, we propose using a differentiable proxy objective, for instance, the supervised fine-tuning (SFT) loss for question-answering tasks, which provides a smooth surrogate while remaining aligned with the discrete evaluation metric. This approach has proven to be highly effective in practice.

**Long outer-loop horizons.** Another challenge arises when the outer-loop duration parameter $n_2$ is greater than one. In this case, computing the gradient of the mixture weights becomes intractable. Without constraints on $n_2$, one would either need to rely on built-in mechanisms in PyTorch (Paszke et al., 2019), such as backpropagation-through-time (BPTT), which quickly becomes prohibitively memory-intensive in large-model settings, or fall back on general gradient-estimation techniques such as finite differences, which again are slow and unstable. To avoid these pitfalls, we restrict $n_2 = 1$ whenever possible, which not only yields a closed-form gradient but also delivers the most stable and efficient optimization behavior.

## 4 Experiments

To comprehensively evaluate the effectiveness of our proposed framework, we conduct experiments on data mixture optimization across different stages of large language model (LLM) training, including both pre-training and post-training. The compared methods cover a wide spectrum of approaches, ranging from human expert tuning to proxy-based search methods such as DoReMi (Xie et al., 2024), RegMix (Liu et al., 2024) and CLIMB (Diao et al., 2025). The subsequent sections are organized as follows: Section 4.1 presents results on pre-training mixture optimization. Section 4.2 reports experiments in post-training settings. Section 4.3 provides ablation studies.

### 4.1 Pre-training Stage Experiments

**Setups.** Following prior work (Liu et al., 2024), we conduct our experiments on the Pile dataset (Gao et al., 2020), focusing on the 17 uncopyrighted subsets available on HuggingFace. For mixture optimization in the pre-training stage, we employ small proxy models (e.g., 1M parameters) trained on up to 1B tokens. To test the method's generalization ability, consistent with Liu et al. (2024), we use the loss on a representative and diverse part of the training data (the Pile-cc sub-set (Gao et al., 2020)) as the search target. For FASTMIX, we employ only a single proxy model, whereas RegMix uses 512 by following (Liu et al., 2024) proxy models and CLIMB uses 64 (Diao et al., 2025). For the Human Heuristic baseline, we directly adopt the manually tuned mixture configuration reported in (Liu et al., 2024) to ensure fairness. After the search stage, we use the mixture configurations obtained by each method to train a 1B-parameter model on 25B tokens. For evaluation, we focus on the accuracy of the pretrained model on a suite of downstream task benchmarks, including Social IQA (Sap et al., 2019), HellaSwag (Zellers et al., 2019), PiQA (Bisk et al., 2020), et.al. In addition, we also examine the time cost incurred by different methods during the search stage.

**Results.** As shown in Figure 2, our proposed method, FASTMIX, demonstrates significant advantages in both downstream task performance and computational efficiency compared to existing data mixture strategies. It achieves the highest average performance score of 48.2 and the best average rank of 1 across all 14 downstream benchmarks, outperforming strong baselines including CLIMB (47.5) and RegMix (47.2). This top ranking underscores its consistent and robust generalization capabilities, further evidenced by its leading results on 9 of the 14 individual tasks. Most notably, FASTMIX offers a dramatic improvement in search efficiency, requiring only 1.3 GPU-hours to

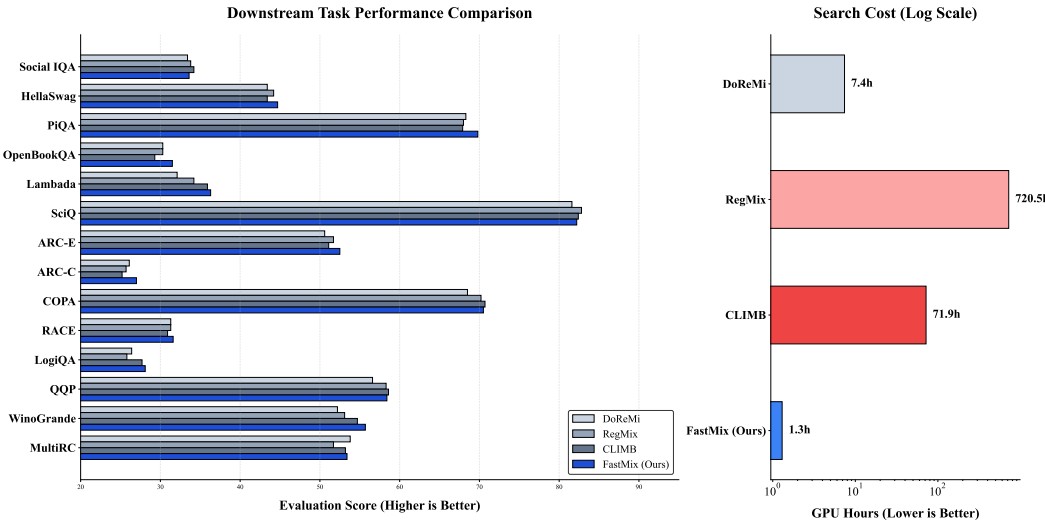

Figure 2: Comparative evaluation of different data mixture strategies in the context of large-scale pretraining, examining their impact on both downstream task performance and training efficiency.

identify the optimal mixture. This is orders of magnitude faster than other automated methods, such as CLIMB (71.9 GPU-hours) and RegMix (720.5 GPU-hours), validating the efficacy of our single proxy model and gradient-based optimization approach. Collectively, these results confirm that FASTMIX not only discovers superior data mixture configurations but also drastically reduces the computational overhead of the search process, offering a scalable and practical solution for large-scale model training.

## 4.2 POST-TRAINING STAGE EXPERIMENTS

**Setups.** Building on our pre-training success, we next validated FASTMIX in the post-training stage, aiming to optimize data mixtures for specialized tasks on the Qwen2.5-Math-Instruct 7B model (Hui et al., 2024). For this study, we sourced supervised fine-tuning (SFT) data from eight distinct domains, including Math (OpenR1-Math-220k (Open-R1 Team, 2024)), Code (the programming-related subset from the OpenThoughts-114K (Guha et al., 2025)), Dialogue (ShareGPT (RyokoAI, 2023)), and STEM (Platypus (Lee et al., 2023)). Our optimization search objective was a 1:1 weighted sum of scores from two mathematical benchmarks, the simpler GSM8K (Cobbe et al., 2021) and the more challenging gaokao2023en (MARIO-Math-Reasoning, 2023). To evaluate the model's generalization capabilities, we extended our test suite beyond math (MATH (Hendrycks et al., 2021), AIME-24 (Jia, 2024)) to include tasks in coding (LiveCodeBench-v2 (Jiang et al., 2024)) and STEM question-answering (GPQA-Diamond (Rein et al., 2023)). A significant challenge in the post-training setting is the absence of very small (e.g., 10M parameter) proxy models. Therefore, we had to conduct our search using proxy models of approximately 1 billion parameters (Qwen2.5-1.5B-Instruct (Qwen et al., 2025)), with evaluation performed on larger models (7B). This constraint exposed a critical limitation of resource-intensive methods (Liu et al., 2024; Diao et al., 2025), which require training hundreds of proxy models. Given the immense computational cost, our cluster was unable to support hundreds of full 1B-model training runs, so we had to reduce the number of proxy models for both RegMix and CLIMB to just 64. In contrast, FASTMIX's reliance on a single proxy model enabled it to operate efficiently within these resource limitations, highlighting its superior scalability for larger-scale tasks.

**Results.** In the post-training (SFT) stage, the advantages of FASTMIX are further solidified, demonstrating an even more dominant performance as shown in Figure 3. Our method achieved the highest score across all four benchmarks spanning mathematics, coding, and general question-answering, resulting in a superior average performance of 65.4 and a top rank of 1, by a significant 5.5 point lead over the next best method, CLIMB (59.9) (Diao et al., 2025). Crucially, these results highlight the exceptional generalization capability of FASTMIX. While all automated methods used performance on mathematics benchmarks (GSM8K and gaokao2023en) as the guidance signal for opti-

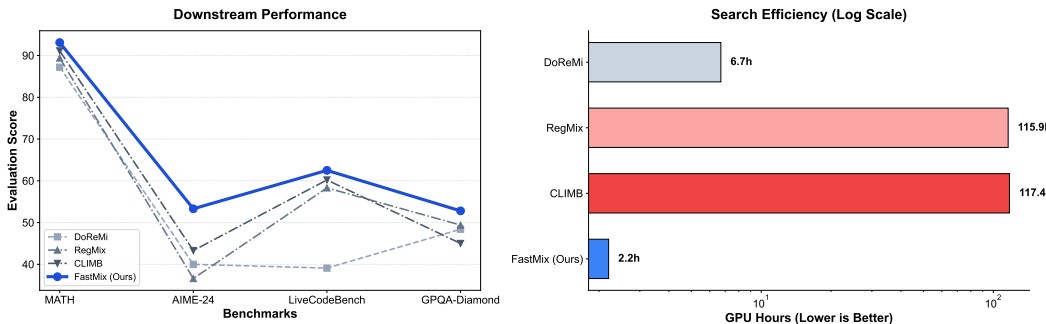

Figure 3: Comparative evaluation of different data mixture strategies in the context of large-scale post-training (SFT), examining the efficiency and downstream task performance.

mization, FASTMIX not only excelled in the math domain but also achieved the best performance on LiveCodeBench (coding) and GPQA-Diamond (STEM QA). This strongly indicates that the data mixture identified by FASTMIX avoids overfitting to the optimization signal and instead fosters a more fundamental and comprehensive improvement in the model's capabilities, all while maintaining remarkable efficiency by completing its search in just 2.2 GPU hours, substantially faster than RegMix (115.9 hours) and CLIMB (117.4 hours).

### 4.3 ABLATION STUDY

This section presents a comprehensive ablation study to validate the key design choices of our proposed FASTMIX framework. All experiments were conducted in a pre-training setting, and the reported results are the average accuracy across 14 benchmarks.

#### 4.3.1 ABLATION FOR THE LOOP DURATION HYPER-PARAMETERS $n_1$ AND $n_2$.

The hyper-parameter $n_1$ represents the **inner-loop duration**, which is the number of steps the model's parameters are updated for each single update of the mixture weights. We ablated the **inner-loop duration** ($n_1$) to find the optimal update frequency for the model parameters in Fig.4(a). Performance rose from 47.3 to a peak of 48.2 as $n_1$ increased from 1 to 20. The score then plateaued and began to decline beyond $n_1 = 40$, suggesting that a moderately long inner loop is needed to effectively learn from the mixture, while an overly long loop can lead to an insufficient searching process.

The hyper-parameter $n_2$ represents the **outer-loop duration**, which determines how many inner-loop steps occur before the mixture weights are updated based on the validation loss. Notably, we can derive a closed-form solution for the mixture weight gradient only when $n_2 = 1$. For cases where $n_2 \neq 1$, we must rely on general-purpose, but often inefficient, methods like finite-difference algorithms, which we did not consider for our primary approach. Our experiments confirmed this design choice in Fig.4(a). We tested $n_2$ values of $\{1, 10, 20, 40\}$, and the average performance was highest with $n_2 = 1$ at 48.1. Performance consistently decreased as $n_2$ increased, falling to 44.2 for $n_2 = 40$. These results demonstrate that the efficiency and numerical stability of our closed-form gradient are not only computationally beneficial but also lead to superior empirical performance.

#### 4.3.2 ABLATION FOR THE REGULARIZATION TERMS.

First, we investigated the effect of the **entropy regularization coefficient** $\lambda$. This term encourages a diverse data mixture, preventing the model from focusing too heavily on a single data source. We varied $\lambda$ across a wide range, from $0.1$ down to $10^{-7}$. As illustrated in Fig.4(b), the algorithm's performance is quite robust when $\lambda$ is less than $10^{-5}$. However, using an excessively large $\lambda$ (e.g., $1.0$) significantly disrupts the optimization process, causing the search to fail to converge on a high-performing solution. This suggests that while some degree of regularization is beneficial, a strong entropy penalty can prevent the algorithm from finding the truly optimal, and potentially non-uniform, mixture.

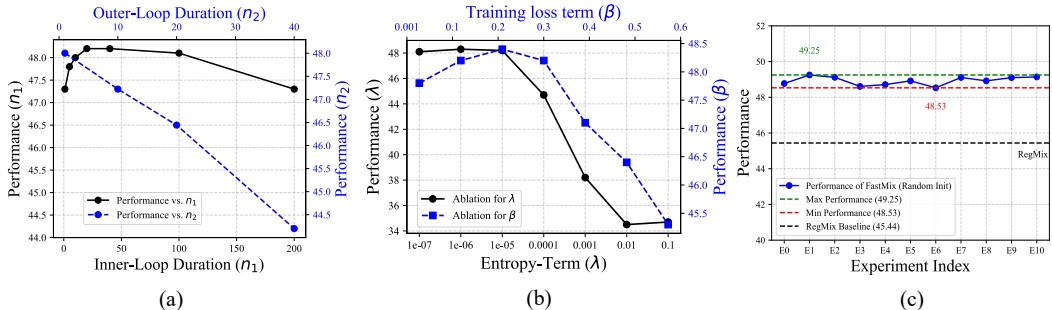

Figure 4: This figure presents a series of ablation studies to validate the design choices of our proposed FASTMIX framework. (a) An analysis of the inner-loop ($n_1$) and outer-loop ($n_2$) durations. (b) The effect of varying the entropy ($\lambda$) and auxiliary training loss ($\beta$) coefficients. (c) A comparison of the final search performance across multiple random initializations.

Next, we ablated the coefficient for the **auxiliary training loss** $\beta$, which helps guide the search towards solutions that generalize well. We tested $\beta$ values from 0.001 to 0.6. The results indicate that a moderate value for $\beta$ is optimal. Performance was highest when $\beta$ was between 0.1 and 0.3, suggesting that a balanced signal from both the validation and training losses is crucial for finding an effective data mixture. Values that are too small provide insufficient regularization, while values that are too large might cause the model to over-optimize for the training distribution at the expense of generalization. The trends for both $\lambda$ and $\beta$ are clearly visualized in Fig.4(b).

### 4.3.3    ABLATION FOR THE INITIALIZATION.

We next investigated the impact of the initial solution on the search results for FASTMIX. We conducted several experiments with different random initializations, and the results are presented in Fig.4(c). As shown in the figure, our method demonstrates remarkable stability. Across 11 different random initializations (labeled E0-E10), the final performance consistently remained high, with an average score of 48.34 and a standard deviation of only 0.48. The results showed a tight clustering, with the highest score at 49.25 and the lowest at 48.53. This consistency suggests that FASTMIX's optimization process is robust and largely independent of its starting point. In all cases, the performance significantly surpassed the baseline RegMix method, which achieved a score of 45.44. This robustness confirms that our gradient-based search effectively navigates the optimization landscape to find high-quality solutions, regardless of the initial conditions.

## 5    CONCLUSION

We introduced FASTMIX, an efficient framework for discovering data mixtures for large-model training. Our key contribution is a weighted bilevel reformulation of mixture selection: via a reparameterization, optimizing sampling ratios becomes equivalent to learning per-source loss weights, enabling mixture coefficients to be differentiable. This permits joint, gradient-based optimization of both the model and the mixture using a single proxy model rather than hundreds. Across pre-training and post-training, FastMix delivers superior accuracy with orders-of-magnitude lower search cost, making data mixture optimization practical, scalable, and robust for next-generation LLMs.

## 6    ACKNOWLEDGMENT

The work has been supported by Hong Kong Research Grant Council-General Research Fund Scheme (Grant No. 17202422, 17212923, 17215025), Theme-based Research (Grant No. T45-701/22-R), and Strategic Topics Grant (Grant No. STG3/E-605/25-N). Part of the described research work is conducted in the JC STEM Lab of Robotics for Soft Materials funded by The Hong Kong Jockey Club Charities Trust.

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

LIMITATIONS AND FUTURE WORK.

Our current FASTMIX framework focuses on optimizing a static data mixture that remains fixed throughout the training process. While this approach has proven highly effective, a more dynamic mixture strategy could offer further improvements. Future work will explore methods for optimizing a data mixture that adapts over time, such as by incorporating curriculum learning principles. This would allow the model to learn from different data sources at various stages of training, potentially accelerating convergence and enhancing performance.

ETHICS STATEMENT

This research adheres to the ICLR Code of Ethics. The primary contribution of our work is to enhance the efficiency of the large model training process, which directly leads to a reduction in energy consumption and computational resources. By making AI development more sustainable, we aim to make a positive contribution to society. We acknowledge the importance of the responsible application of this technology. Our research does not involve the collection or use of any new personally identifiable information, and all experiments were conducted on publicly available datasets.

LLM-USAGE STATEMENT

The authors used a large language model to assist with language polishing, grammar correction, and typo identification in this paper. The ideas, methodology, experimental design, and results presented are the sole work of the authors.

BROADER IMPACT

The development of FASTMIX has significant positive implications for the large model training community. By automating and accelerating the data mixture optimization process, our framework drastically reduces the computational resources and human effort required for this critical task. This increased efficiency translates directly into reduced energy consumption and a lower carbon footprint, promoting more environmentally sustainable AI research and development.

