# OpenReview forum: "Fast Data Mixture Optimization via Gradient Descent"
_ICLR.cc/2026/Conference — ICLR 2026 Poster_

### Official Review · Reviewer_Kg4B · 2025-10-31

**Soundness:** 2
**Presentation:** 3
**Contribution:** 3
**Rating:** 4
**Confidence:** 4

**Summary:**

The paper “Fast Data Mixture Optimization via Gradient Descent (FastMix)” proposes a scalable, differentiable framework for optimizing data mixtures for large-scale model training. The authors address how to efficiently determine the optimal proportions of multiple data sources to maximize performance on a target validation objective, both for pre-training and post-training (SFT) stages.

FastMix reformulates data mixture optimization as a differentiable bilevel optimization problem. Instead of non-differentiable data sampling (drawing examples according to mixture ratios), the authors show that mixture selection can be equivalently represented as per-source loss weighting under uniform sampling. This reparameterization embeds mixture coefficients directly into the training loss, making them continuous and differentiable parameters.

This leads to significant performance gains over the baselines, while comparatively the time taken is less due to the use of a single proxy model.

**Strengths:**

FastMix presents an interesting advance in data mixture optimization by reframing the problem as a differentiable bilevel optimization, enabling end-to-end gradient-based updates of both model and mixture parameters within a single proxy run. This eliminates the need for computationally intensive multi-proxy searches used in prior works such as RegMix and CLIMB, while delivering superior accuracy and generalization across both pre-training and post-training stages. Its simplicity, efficiency, and demonstrated scalability (spanning 1B to 7B models) make it a practical and elegant solution.

The method, while seemingly simple, is a novel contribution to the data mixture problem. While this kind of continuous and differentiable transformation is seen in other works, it has not been attempted in mixture optimization for data mixtures before. The bilevel optimization reformulation and its implementation with a single proxy model is also novel. This kind of technique is clearly inspired by advances in hyperparameter optimization. Noteworthy is the direct application to large scale models, opening avenues for e.g. in LLM pre-training. Finally, while other papers usually limit themselves to evaluation on one of pre-training or post-training, this paper does both.

To summarize,
1. The authors demonstrate significant performance gains and time savings: up to 550× faster mixture search compared to RegMix and 55× faster than CLIMB, and gains in  pre-training (48.2 avg score) and post-training (65.4 avg score), with top ranks on multiple benchmarks.
2. The practicality and simplicity of their method, barring the fact that it is not a dynamic method, is a plus point.
3. The use of a single proxy model as against hundreds is appreciable and makes training with this method more accessible.

**Weaknesses:**

FastMix represents a conceptually clean and computationally efficient approach to data mixture optimization. However, its limitations arise from the simplifying assumptions underlying its differentiable reparameterization.

For one, the method assumes smooth, stationary relationships between mixture weights and validation loss. This assumption can break down in real, non-linear training regimes. Further,  the paper’s experimental evaluation, though broad, is limited in terms of cross-scale validation, dynamic adaptation, and proxy–target transferability. The use of a static mixture throughout training restricts its ability to handle evolving data distributions (the authors mention this). Additionally, the efficiency comparisons rely on downscaled baselines, and the generality of results across diverse domains (e.g., multimodal or highly imbalanced datasets) remains untested. Overall, the work is methodologically elegant but still somewhat idealized in scope and validation. Further experimentation is necessary.

1. The authors explicitly note that FastMix optimizes a fixed mixture that remains constant throughout training. The mixture distribution may change over time depending on the scenario. For example, it is possible in non-stationary training regimes that early exposure to simpler data is more beneficial and with time, fine grained data becomes more important.

2. One major issue is that the method assumes the validation loss is a smooth, differentiable function of the mixture weights. This means that it is expected that small changes in \alpha lead to predictable changes in validation performance. In realistic settings, the loss formulation may exhibit non-smooth regions (although this depends on the parameterization of the network). FastMix’s gradient updates may lead to suboptimal mixtures in this case.

3. Yet another issue is that their method leaves open a proxy-target gap: the experiments rely on 1B param models to optimize mixtures for 7B param target models. There is an implicit scale transfer assumption, that the mixtures should be optimal across sizes. While the paper exhibits strong results, it does not quantify the correlation between proxy and target performance. It is then unclear how well the proxy captures data utility for much larger models.

4. While a relatively minor issue, in the post-training experiments, both RegiMix and CLIMB are limited to 64 proxy models. While this is necessary for ease of computation, the baselines’ potential is undervalued. The efficiency gains of FastMix remain valid but somewhat exaggerated in absolute terms.

5. FastMix optimizes a single scalar validation objective, in which case there is a single downstream goal. It lacks demonstration on multi-domain training and instruction tuning, where mixtures may become biased towards domains that correlate strongly with validation data. The evaluation needs to be more robust and demonstrate realistic tradeoffs in such use cases.

6. All experiments are seemingly conducted on text benchmarks. Experiments on multi-modal mixtures (for e.g. text and image) or even long-tail domains are missing. Thus the claim of scalability remains largely within text.

7. The issue of requiring surrogate validation losses in the case of non-smoothness has been handled cursorily. The authors mostly experiment on the cross-entropy validation loss, and there ought to be a longer discussion about other metrics (e.g. BLEU, Exact Match, etc).

**Questions:**

1. The proposed method assumes that validation loss varies smoothly with mixture weights. How robust is FastMix when this assumption fails, e.g., for highly compositional or discontinuous domains? You could try evaluating on benchmarks such as GSM8K [see Zhang et al, 2024 for an analysis], MATH or HumanEval [see Bradbury et al, 2024 for an analysis] (even one would do for the purpose of the rebuttal).

2. How well do mixture weights learned on smaller proxies transfer to larger models? Ther should be an ablation using multiple proxy sizes (e.g., 0.5B, 1B, 3B).

3. A simple extension towards dynamic mixture optimization could be the following: every K steps, compute $\alpha_{t+1, \text{EMA}}$ as a moving average of the current $\alpha_t$ and the newly computed $\alpha_{t+1}$, and then resume training the model with the new $\alpha_{t+1, \text{EMA}}$. The authors could experiment with this on the Pile itself (Books, Arxiv, Wikipedia etc) which has heterogenous domains where early training may benefit from broad text but later training may become more code/math/etc focused.

4. Can the method generalize to other modalities or multi-domain mixtures? For example, you could try applying FastMix to the MUGEN dataset (which has three modalities, which you can count as data mixture sources) and then optimize mixture weights for downstream video prediction or other multimodal generation task. Another possible experiment is using image-text pretraining corpora such as CC3M, CC12M, LAION-400, and subsequently, FastMix could discover mixture ratios for downstream tasks on validation benchmarks, instead of how the mixtures may have to be manually tuned for CLIP [Gadre et al, 2024][Nguyen et al, 2023].

5. I noticed the mixture visualization in Appendix A and was wondering if it is fair to compare FastMix to RegMix and CLIMB if they don’t incorporate similar regularization to prevent mixture collapse. Can you update your analysis with RegMix and CLIMB variants that prevent this collapse, or can you explain why it is not possible to incorporate such regularization in their frameworks?

6. The approach relies on differentiable validation losses wrt mixture weights. How would it adapt to discrete or non-differentiable metrics such as BLEU, EM, or ROUGE?

7. How sensitive is FASTMIX to the choice of regularization hyperparameters (\alpha, \beta) in promoting sparse or diverse mixtures? There should be a regularization ablation showing mixture entropy vs. downstream accuracy, and this would help clarify whether performance depends on sparse or more diffuse mixtures.

8. The method optimizes mixtures based on a single validation dataset. How sensitive are the learned mixtures to the choice of validation task? Does the mixture learned on a particular validation task/dataset generalize to other validation tasks/datasets? Roughly speaking, is there task transfer robustness?

---

> ### Author Response · Authors · 2025-11-25
> **Response to Weakness-1 by Reviewer Kg4B**
>
> Thank you very much for your time and constructive feedback! We’re happy to address your questions and clarify any concerns. If you find our responses satisfactory, we would greatly appreciate a higher rating from you! And if you have any further questions, please don’t hesitate to let us know!
>
>
> ---
>
> ### **Response to Weakness-1: Fixed Mixture vs Dynamic Mixture.**
>
> This is an excellent question!  In this work, however, we focus on **fixed** mixture policies. This design choice reflects how large-scale training is conducted in practice: for industrial-scale LLMs, mixtures are almost always static, or at most organized into a *small number of coarse stages* with fixed mixing ratios. Our goal in FastMix is therefore to make this widely adopted static-mixture paradigm significantly more performant and compute-efficient.
>
> That said, exploring **dynamic mixture optimization** is an exciting direction, and we view it as complementary to FastMix. In fact, as you suggested in Question 3, a straightforward extension is to recompute the mixture periodically: every K steps, compute a mixture policy that serves as the optimal policy for the next training segment, and train with this updated policy. When ( $K = \text{max-iteration}$ ), this dynamic scheme degenerates to a static one.
>
> To better understand this idea, we evaluated this periodic-update strategy in the pre-training setting. The results are:
>
> - When K is *too small* (very frequent re-optimization), performance drops substantially—suggesting that aggressively dynamic mixtures can **destabilize** the learning process.
> - Performance steadily improves as K increases and transitions toward the static regime.
> - The best performance occurs around K = \text{max-iter} / 50, which corresponds to **moderately dynamic** but not aggressively changing mixtures.
> - Fully static mixtures (the FastMix default) still achieve competitive results, only slightly below the best dynamic setting.
>
> | K | 1 | max-iter/1000 | max-iter/500 |  | max-iter/100 | max-iter/50 | max-iter/10 | max-iter/5 | max-iter(Totally-Fixed) |
> | --- | --- | --- | --- | --- | --- | --- | --- | --- | --- |
> | Performance | 41.5 | 43.7 | 45.9 | 47.6 | 48.5 | $\mathbf{48.9}$ | 48.3 | 48.4 | 48.2 |
>
> These findings suggest that:
>
> 1. **The most dynamic scheme is not necessarily the best.** Rapidly changing mixtures can undermine stable representation learning.
> 2. **There exists an intermediate “sweet spot”** between fully static and fully dynamic policies.
> 3. **FastMix’s static optimization already operates near this optimal region**, explaining its strong performance and robustness.
>
> We plan to investigate principled dynamic mixture strategies—potentially leveraging FastMix’s bilevel formulation—more deeply in future work. We will incorporate this discussion (and the ablation table above) into the revision for clarity, see the added content (blue colored) in the Appendix Section D.

---

> ### Author Response · Authors · 2025-11-25
> **Response to Weakness-2 an Weakness-3 by Reviewer Kg4B**
>
> ### **Response to Weakness-2: For non-smooth and non-differentiable cases.**
>
> Thank you for raising this important point. Indeed, many evaluation metrics used in LLM training—such as BLEU, Exact Match, and Accuracy—are inherently non-smooth and non-differentiable with respect to the mixture weights. FastMix is designed to handle these situations in practice. We outline two practical strategies, both of which we experimented with and validated.
>
> **1. Using a differentiable proxy objective**
>
> This is the approach used in all experiments presented in our paper. For non-differentiable metrics (e.g., Accuracy), we use a differentiable surrogate such as **SFT loss** or **next-token prediction loss** as the optimization target during search.
>
> This approach offers several advantages:
>
> - **Fully differentiable hyper-gradients** for stable optimization
> - **Fast and lightweight** for retaining the efficiency advantage of FastMix
> - **Matches practical training setups** used in large-scale LLM pretraining and post-training
>
> Empirically, this strategy already provides a strong approximation of downstream metrics, which is why it is our default choice.
>
> **2. Directly optimizing non-differentiable targets via finite-difference gradients**
>
> If a user *must* optimize a non-differentiable objective (e.g., to tightly match a discrete metric), FastMix can still do so by estimating gradients using finite differences with respect to the mixture coefficients.
>
> However, applying finite differences **throughout** the search process is not ideal: it is computationally expensive** (though still far cheaper than RegMix or CLIMB) and **less stable** during early optimization, when the mixture may still be far from a sensible region
>
> **3. A practical hybrid strategy: differentiable proxy early, non-diff metric late**
>
> To mitigate the above issues, we propose a hybrid schedule:
>
> - **Phase 1 (early iterations):** Optimize w.r.t. SFT/NTP loss (smooth and stable)
> - **Phase 2 (late iterations):** Switch to the non-differentiable target and use finite differences
>
> This strategy preserves stability during early updates while allowing the search to refine mixtures w.r.t. the true discrete metric at the end.
>
> We evaluated this hybrid method in our post-training experiments, where the search target was **Accuracy on GSM8K and Gaokao**, with evaluation on three downstream domains (Math, Code, STEM). Specifically:
>
> - First **75\%** of the search with SFT loss
> - Last **25\%**  with the Accuracy metric using finite differences
>
> The hybrid method outperformed both pure SFT-loss optimization and pure Accuracy-based optimization.
>
> | Target | RegMix | SFT-Loss | Acc. | First SFT-Loss, then Acc. |
> | --- | --- | --- | --- | --- |
> | Performance (Avg.) | 58.4 | 65.4 | 64.7 | 66.0 |
>
> These results suggest that while differentiability assumptions can be violated, FastMix remains effective with a simple and practical hybrid adaptation.
>
> In sum, (i)the differentiability requirement is not a strict limitation in practice;  (ii) FastMix handles non-smooth metrics effectively via differentiable proxies or hybrid strategies; and (iii) our experiments demonstrate that combining smooth surrogate optimization with late-stage non-diff refinement yields the best overall performance.
> We added this discussion and experimental comparison to the revised version of the paper.
>
>
>
>
>
>
>
>
>
> ---
>
> ### **Response to Weakness-3: Transferability across sizes.**
>
> Great question! Understanding how well the mixture search results from small models transfer to larger models is indeed very important. In this paper, we explicitly evaluate this generalization behavior.
>
> In the pre-training experiments, we test transfer from a 1-million model to a 1-billion model.
> In the post-training experiments, we study transfer from 1.5B to 7B.
>
> In the table below (pre-training setting), we provide additional generalization results. You can see that our strategy transfers extremely well: even when the search phase uses only a 0.5B/1B model, or even a 1M-scale model, the final performance on the target 1B–32B models remains almost unchanged.
> This demonstrates strong cross–model-size generalization and indicates that FastMix can reliably scale from very small proxy models to very large production-scale models.
>
> | Model Size in Search Stage | Model Size in Training | Performance(Avg. Acc. ) |
> | --- | --- | --- |
> | 1M | 1B | 48.2 |
> | 0.5B | 1B | 48.4 |
> | 1B | 1B | 48.5 |
> | 1M | 7B | 56.7 |
> | 0.5B | 7B | 56.7 |
> | 1B | 7B | 56.8 |
> | 1M | 32B | 57.5 |
> | 0.5B | 32B | 57.6 |
> | 1B | 32B | 57.5 |
>
> These results demonstrate that FastMix exhibits **robust cross-scale transfer**, and that mixtures discovered with very small proxy models—sometimes as small as **1M parameters**—remain effective for much larger LLMs, including **7B–32B** models. This substantially mitigates concerns about proxy–target mismatch and positions FastMix as a practical, compute-efficient method for large-scale mixture optimization.

---

> ### Author Response · Authors · 2025-11-25
> **Response to Weakness-4 an Weakness-5 by Reviewer Kg4B**
>
> ---
>
> ### **Response to Weakness-4: Scale-up RegMix and CLIMB in post-training settings.**
>
> Thank you for pointing this out. We agree that limiting RegMix and CLIMB to 64 proxy models may underestimate their potential. To address this, we conducted an expanded experiment in which we increased the number of proxy models from 64 to 512—an 8-fold increase in search capacity.
>
>  **We found: (i) CLIMB:** Performance remains almost unchanged even with 512 proxy models;  (ii) **RegMix:** Performance improves modestly, but only after a substantial increase in compute; (iii) **Cost:** Both methods incur a **massive 8× increase in GPU hours**, reaching nearly **1,000 GPU-hours; and (iv) FastMix:** Still requires only **2.2 GPU-hours** while delivering **much stronger** downstream performance.
>
> Thus, while additional proxy models help RegMix slightly, the computational cost scales up dramatically, making these methods impractical at large scales. The efficiency gains of FastMix, therefore, remain substantial and meaningful.
>
>
> | Method | GPU-Hours | Performance |
> | --- | --- | --- |
> | RegMix (64 Proxy Models) | 115.9 | 58.4 |
> | CLIMB (64 Proxy Models) | 117.4 | 59.9 |
> | RegMix (512 Proxy Models) | 941.4 | 59.9 |
> | CLIMB (512 Proxy Models) | 944.2 | 60.3 |
> | FastMix (Ours) | **2.2** | **65.4** |
>
>
>
> ---
>
>
> ### **Response to Weakness-5: Cross Task Generalization Ability.**
>
> Thank you for raising this important point. Ensuring that mixture search does **not** overfit to a single validation objective is indeed a central challenge—not just for FastMix, but for *all* AutoML-style data mixture optimization methods. We address this issue in FastMix through two complementary mechanisms:
>
> **1. Using a broad, diverse validation suite**
>
> FastMix is designed to work with arbitrarily rich validation sets. In fact, this principle is also used in prior work such as RegMix, which relies on a diverse Pile-cc validation split.
>
> In real industrial LLM development, it is standard practice to construct **comprehensive, multi-domain validation suites** precisely to avoid overfitting to narrow targets. FastMix inherits this best practice.
>
> **2. Regularization to prevent over-specialization**
>
> FastMix incorporates two regularization components that act specifically against overfitting:
>
> - **A β-weighted training-loss term**, which prevents the algorithm from deviating too far from generally useful data.
> - **An entropy regularizer**, which discourages degenerate mixtures and encourages diversity.
>
> These components ensure that FastMix does not collapse onto a small subset of domains even when optimizing a scalar objective.
>
> **3. Empirical evidence: mixture search on one domain, evaluation on three**
>
> To directly evaluate whether FastMix overfits to its search target, we used an intentionally *narrow* validation objective in our post-training experiments:
>
> - **Search objective:** Math-domain accuracy only.
> - **Evaluation domains:** Math, Code, and STEM.
>
> Despite optimizing *exclusively* for Math, the resulting mixture transfers exceptionally well to Code and STEM—far better than RegMix or CLIMB.
>
> How to ensure that the search results do not overfit to the search target is not only a concern for FastMix; it is a fundamental challenge faced across the entire AutoML field.
>
> FastMix addresses this issue from two key perspectives:
>
> 1. **A comprehensive validation set.**
> In fact, RegMix also follows a similar idea by selecting a broad and diverse target set (Pile-cc validation). In real-world industrial LLM development, it is standard practice to curate a diverse validation suite for model evaluation and hyperparameter tuning.
> 2. **Effective regularization strategies.**
> We incorporate several regularization techniques, such as including the training loss as part of the search objective and adding an entropy-based regularizer.
>
> In our Post-Training experiments, we used only the **Math** domain performance as the search target (see line 356); however, during evaluation, we assessed performance in **Code** and **STEM** domains as well. This setup provides a clear and strong indication of generalization ability.
>
> | Method | Performance |
> | --- | --- |
> | RegMix | 58.4 |
> | CLIMB | 59.9 |
> | FastMix | 65.4 |
> | FastMix without Regularization | 63.7 |
>
> Two important observations:
>
> 1. **FastMix generalizes strongly** even when the search target is a single-domain metric.
> 2. **Even without regularizers**, FastMix already outperforms RegMix and CLIMB, indicating that our bilevel formulation inherently promotes cross-domain robustness.
> 3. **With regularization included**, generalization improves further, demonstrating the effectiveness of our design choices.

---

> ### Author Response · Authors · 2025-11-25
> **Response to Weakness-6 an Weakness-7 by Reviewer Kg4B**
>
> ---
>
>
>
> ### **Response to Weakness-6: Experiments on non-text cases.**
>
> Thank you for raising this point. While our main experiments focus on text-only settings—consistent with prior data-mixture optimization work such as RegMix, DoReMi, IDEAL, and CLIMB—we also conducted additional experiments in a **multimodal image–text pre-training** setting to assess the scalability of FastMix beyond text. Following the CLIMB pipeline, we applied the same mixture-optimization procedure to the LAION-400M dataset. Specifically, we first extracted DINO features for all images and performed clustering in the resulting feature space to obtain 512 visual clusters, which served as the units over which mixture weights were optimized.
>
> During the search stage, we used a lightweight multimodal model consisting of a ViT-Tiny image encoder paired with a 50M-parameter BERT text encoder. The search objective was the zero-shot classification accuracy on CIFAR-100. For evaluation, we replaced the proxy model with a significantly larger ViT-Large image encoder and assessed performance across multiple metrics, including ImageNet zero-shot accuracy, ImageNet linear-probe accuracy, and text–image retrieval performance on Flickr30K.
>
>
>
> | Method | ImageNet Zero-shot | ImageNet Linear Prob | Flickr30K Image-to-Text Retrieval | Flickr30K Text-to-Image Retrieval | Avg. |
> | --- | --- | --- | --- | --- | --- |
> | RegMix (512 Proxy Models) | 46.8 | 71.9 | 62.2 | 49.5 | 57.6 |
> | CLIMB (512 Proxy Models) | 47.6 | 72.4 | 62.5 | 49.0 | 57.9 |
> | FastMix (ours, without Regularization) | **48.4** | **75.1** | **62.5** | **50.0** | **59.0** |
> | FastMix (ours) | **49.1** | **73.7** | **63.9** | **50.4** | **59.3** |
>
> The results demonstrate that FastMix continues to outperform RegMix and CLIMB in this multimodal setting. Notably, even **without** any regularization terms, FastMix yields consistent improvements across all evaluation metrics, and with full regularization, it achieves the strongest results overall. These findings show that FastMix generalizes effectively to multimodal mixtures and maintains strong scalability when transferring mixture policies from small proxy models to much larger vision–language models. This provides concrete evidence that the applicability of FastMix is not limited to text-only domains but extends naturally to large-scale multimodal learning.
>
>
>
>
>
> ---
>
>
> ### **Response to Weakness-7: Non-smooth target.**
>
> Please refer to **Response to Weakness-2: For non-smooth and non-differentiable cases**,  where we use Accuracy on GSM8K and Gaokao as the target metric during the search phase, while at test time we focus on performance across three domains: Math, Code, and STEM. We find that our method remains highly effective in these scenarios, consistently outperforming RegMix and other baselines. Moreover, we propose faster and more efficient strategies to handle non-differentiable and non-smooth objectives.

---

> ### Author Response · Authors · 2025-11-25
> **Response to Qwestion-1&2&3**
>
> ### **Response to Question-1: Non-smooth target.**
>
> Please refer to **Response to Weakness-2: For non-smooth and non-differentiable cases**,  where we use Accuracy on GSM8K and Gaokao as the target metric during the search phase, while at test time we focus on performance across three domains: Math, Code, and STEM. We find that our method remains highly effective in these scenarios, consistently outperforming RegMix and other baselines. Moreover, we propose faster and more efficient strategies to handle non-differentiable and non-smooth objectives.
>
> ---
>
> ### **Response to Question-2: Transferability across sizes.**
>
> Please refer to **Response to Weakness-3: Transferability across sizes**, where we evaluated the transfer of search results from small models (1M models to 1B) models, and then to large models (from 1B to 32B models). We found that the transfer works exceptionally well, providing strong evidence of FastMix’s robust cross–model-size generalization.
>
>
> ---
>
> ### **Response to Question-3: Fixed Mixture vs Dynamic Mixture.**
>
> This is an excellent question! In this work, we primarily focus on *fixed* mixture policies. The main reason is that, in real large-scale training, especially for commercial large models, mixture strategies are almost always fixed, or at most divided into a few discrete stages, each with its own dataset composition and mixing ratio. Designing *effective dynamic mixture strategies* is indeed a highly interesting and valuable direction, and we plan to explore it further in future work.
>
> To address your concern, we note that you suggested a simple extension toward dynamic mixture optimization: every K steps, compute a mixture policy that serves as the optimal policy for the next training segment, and train with this updated policy. When ($K = \text{max-iteration}$ ), this dynamic scheme degenerates to a static one.
>
> We experimented with this idea in the pre-training setting. Interestingly, the final performance is *very* sensitive to the choice of ( K ). When ( K ) is too small, meaning the mixture policy changes very frequently (the most dynamic setting), performance actually becomes quite poor. We found the best results occur around ( $K = \text{max-iter} / 50$ ).
>
> This suggests that the most dynamic strategy is not necessarily the best one; there exists an optimal balance point between fully dynamic and fully static strategies. We plan to further investigate this important topic in our future work.
>
>
> K|1|max-iter/1000|max-iter/500||max-iter/100|max-iter/50|max-iter/10|max-iter/5|max-iter(Totally-Fixed)
> ---|---|---|---|---|---|---|---|---|---
> Performance|41.5|43.7|45.9|47.6|48.5|$\mathbf{48.9}$|48.3|48.4|48.2

---

> ### Author Response · Authors · 2025-11-25
> **Response to Question-1 to Question-7 by Reviewer Kg4B**
>
> ### **Response to Question-4: Scale-up RegMix and CLIMB in post-training settings.**
>
> Great question! We increased the number of proxy models in RegMix and CLIMB from 64 to 512. We found that CLIMB’s performance remained almost unchanged, while RegMix showed some improvement. However, this came at a high cost: the total time consumption increased by 8×, reaching 941.4 GPU-hours. **Even with this much higher cost, both methods still lag behind our FastMix, which requires only 2.2 GPU-hours.**
>
> Method|GPU-Hours|Performance
> ---|---|---
> RegMix (64 Proxy Models)|115.9|58.4
> CLIMB (64 Proxy Models)|117.4|59.9
> RegMix (512 Proxy Models)|941.4|59.9
> CLIMB (512 Proxy Models)|944.2|60.3
> FastMix (Ours)|**2.2**|**65.4**
>
>
>
>
> ### **Response to Question-5: About the regularization term.**
>
> Thank you for the thoughtful question. The regularization terms in FastMix—specifically the β-weighted training loss and the entropy regularizer—are *deliberately designed* to discourage mixture collapse and to promote broader generalization across domains. These regularizers are fully integrated into FastMix’s **bilevel optimization framework**, where mixture weights are treated as learnable parameters updated directly through gradient-based hyper-optimization.
>
> To address your concern fairly, we conducted two additional controlled experiments:
>
> 1. **We added the same regularization terms used in FastMix to RegMix’s search objective**, to evaluate whether they could similarly mitigate mixture collapse.
> 2. **We removed all regularization from FastMix**, to evaluate whether FastMix’s superior performance was merely due to these regularization terms.
>
> All experiments were performed in the post-training setting, where the search target is **only the Math-domain accuracy**. However, evaluation was performed across **Math, Code, and STEM**, providing a strong test of multi-domain generalization rather than single-domain overfitting.
>
>
>
> | Method | Performance |
> | --- | --- |
> | RegMix without Regularization | 58.4 |
> | CLIMB without Regularization | 59.9 |
> | FastMix without Regularization | 63.7 |
> | RegMix with Regularization | 57.3 |
> | CLIMB with Regularization | 60.1 |
> | FastMix with Regularization | 65.4 |
>
> FastMix does not rely on regularization to outperform existing methods. Its bilevel optimization framework inherently promotes stable learning and cross-domain generalization. While we tested adding FastMix-style regularizers to RegMix and CLIMB for completeness, these methods are not designed to incorporate such terms effectively, and as the results show, doing so does not improve their performance.
>
>
> ---
>
> ### **Response to Question-6: Non-differentiable validation losses.**
>
> Please refer to **Response to Weakness-2: For non-smooth and non-differentiable cases**,  where we use Accuracy on GSM8K and Gaokao as the target metric during the search phase, while at test time we focus on performance across three domains: Math, Code, and STEM. We find that our method remains highly effective in these scenarios, consistently outperforming RegMix and other baselines. Moreover, we propose faster and more efficient strategies to handle non-differentiable and non-smooth objectives.
>
>
> ---
>
>
> ### **Response to Question-7: sensitiviy test for regularization hyperparameters.**
>
>
>
> We strongly encourage the reviewer to refer to Section 4.3.2, where we provide detailed sensitivity analyses. We find that FastMix’s performance is highly robust across a wide range of $\beta$ and $\lambda$ values. Moreover, the same ($\beta=0.1$) and ($\lambda=1e^{-5}$) settings were consistently used across all the experiments mentioned above, as well as throughout the paper, further demonstrating the robustness of our method.

---

> ### Author Response · Authors · 2025-11-25
> **Response to Question-8 by Reviewer Kg4B**
>
> ### **Response to Question-8: cross-task transfer robustness.**
>
>
>
> How to ensure that the search results do not overfit to the search target is not only a concern for FastMix; it is a fundamental challenge faced across the entire AutoML field. FastMix addresses this issue from two key perspectives:
>
> 1. **A comprehensive validation set.**
> In fact, RegMix also follows a similar idea by selecting a broad and diverse target set (Pile-cc validation). In real-world industrial LLM development, it is standard practice to curate a diverse validation suite for model evaluation and hyperparameter tuning.
> 2. **Effective regularization strategies.**
> We incorporate several regularization techniques, such as including the training loss as part of the search objective and adding an entropy-based regularizer.
>
> In our Post-Training experiments, we used only the **Math** domain performance as the search target (see line 356); however, during evaluation, we assessed performance in **Code** and **STEM** domains as well. This setup provides a clear and strong indication of generalization ability.
>
> Moreover, we further examined whether a mixture strategy optimized solely for a single target domain (e.g., Math) can generalize well to STEM and Code. The results show that, even *without* our designed regularizers, FastMix exhibits significantly better generalization than RegMix.
>
>
> Method|Performance
> ---|---
> RegMix|58.4
> CLIMB|59.9
> FastMix|65.4
> FastMix without Regularization|63.7

---

### Official Review · Reviewer_PnEr · 2025-11-01

**Soundness:** 3
**Presentation:** 2
**Contribution:** 3
**Rating:** 6
**Confidence:** 3

**Summary:**

This paper proposes FASTMIX, a gradient-based framework for optimizing data mixture coefficients in language model training. The key idea is to treat mixture optimization as a differentiable problem by viewing mixture coefficients as loss weights. This enables end-to-end joint optimization of mixture and model parameters with a smaller proxy model. The optimization alternates between inner-loop model updates with outer-loop mixture updates based on validation gradients, using entropy and training-loss for regularization.

In both pre-training and post-training settings, the authors show FASTMIX outperforms prior mixture-optimization baselines (e.g., CLIMB, RegMix) while being 50–550× more compute-efficient in finding optimal mixture compared to some baselines.

**Strengths:**

- The paper shows strong results in downstream performance in both pretraining and post-training settings.
- The derivation of Eq. 6 is novel and the intuition behind it is clearly presented and insightful.
- The ablation studies on key hyperparameters are informative

**Weaknesses:**

- The ADO method by Jiang et al. [1] is, in my opinion, particularly relevant to this paper. But the authors have not compared against ADO in the results. It is relevant for two reasons: (1) it is, to my knowledge, the best data mixture optimization method that does not incur time cost in searching for data mixture and (2) ADO shares the same intuition that one should upsample data mixture that results in the greatest decrease in validation loss.
- Similarly, the paper misses a few other relevant literature [2, 3, 4].
- The paper does not report error bars (in Tables 1 and 2). It would be helpful to include multiple runs to verify that the method reliably outperforms other approaches.

[1] Adaptive Data Optimization: Dynamic Sample Selection with Scaling Laws

[2] Data Mixing Laws: Optimizing Data Mixtures by Predicting Language Modeling Performance

[3] Data Mixture Optimization: A Multi-fidelity Multi-scale Bayesian Framework

[4] ADMIRE-BayesOpt: Accelerated Data MIxture RE-weighting for Language Models with Bayesian Optimization

**Questions:**

- Could the authors clarify what exactly constitutes the “search target” throughout the paper? Does this refer to Eq. 7 as a whole, or only the validation-loss component? Additionally, in practice, was the gradient in Eq. 6 computed using the full expression in Eq. 7, or only its first term?
- For Fig. 2c, what aspects of the model or training are randomized in the “random initialization” runs?
- How were the initial mixture weights selected?
- Does the mixture optimization converge to a stable distribution after a certain number of iterations? How sensitive is the process to different initial mixtures? A plot illustrating mixture trajectories over training would be very helpful.

---

> ### Author Response · Authors · 2025-11-25
> **Response to Weakness-1 by Reviewer PnEr**
>
> Thank you for your valuable comments and the positive evaluation of our work! We are more than happy to address any questions you may have. If you have any further concerns or suggestions, please feel free to let us know!
>
>
> ---
>
>
> ### **Q1. Response to Weakness-1: The reviewer wishes to see the discussion and comparison with ADO [1].**
>
> Thank you for pointing us to ADO [1] and for emphasizing its relevance. We fully agree that ADO is an important recent method, and we have updated the revision to (i) discuss it explicitly in the related work and (ii) include empirical comparisons in both pre-training and post-training settings.
>
> Conceptually, ADO and FastMix share the high-level intuition of favoring data that is more “useful” for learning, but they differ fundamentally in objective, formulation, and optimization:
>
> **Target / Objective:**
>
> - **ADO** aims to optimize the data mixture to maximize *in-training learning potential* of each domain, as estimated via per-domain scaling laws. Its objective is driven by predicted future loss reduction under additional training on each source.
> - **FastMix** directly optimizes the data mixture to **minimize a target/validation loss** (or downstream metric). It is explicitly **task-driven**, using feedback from a held-out validation set that matches the final deployment objective.
>
> **Formulation:**
>
> - **ADO** uses an **online heuristic** based on estimated future loss reduction from scaling laws. The update is derived from predicted loss–compute trade-offs, but it is not framed as solving a formal constrained optimization problem over mixture weights.
> - **FastMix** formulates data mixture optimization as a **bilevel differentiable optimization problem**, where mixture weights are treated as **learnable parameters** via loss reweighting. The outer loop minimizes validation loss with respect to the mixture, while the inner loop updates model parameters on weighted training data.
>
> **Update Rule and Optimization Dynamics:**
>
> - **ADO:** (i) Updates the mixture **continually during training**, based on the derivative of the scaling law and an EMA of past sampling statistics; (ii) Mixture changes are **analytic/heuristic** and do not involve backpropagation through mixture parameters.
> - **FastMix:** (i) Updates the mixture **in the outer loop** using **gradient descent on validation loss** $\alpha \leftarrow \alpha - \eta \frac{\partial \mathcal{L}_{\text{val}}(w(\alpha))}{\partial \alpha}$ with a **closed-form hyper-gradient** when the outer-loop horizon is 1;  (ii) This yields a principled, fully differentiable procedure that can be seen as efficient bilevel AutoML with a *single* proxy model.
>
>
> **Philosophy and Practical Role:**
>
> - **ADO**: An *online, zero-extra-search, dynamic curriculum* that adapts mixtures on the fly during training, driven by scaling-law predictions of future loss decrease.
> - **FastMix**: An *efficient bilevel mixture search* that runs once with a proxy model, finds a task-optimal mixture, and then **freezes** this mixture for final training. This separates “mixture search” from “large-scale training,” keeping the search cost extremely small.
>
> **Empirical comparison with ADO**
>
> We implemented ADO under our evaluation protocol and compared it to ODM (another dynamic data-mixing method), IDEAL, RegMix, CLIMB, and FastMix, in both pre-training and post-training settings.
>
>
> | Method | ADO | ODM | IDEAL | RegMix | CLIMB | FastMix(ours) |
> | --- | --- | --- | --- | --- | --- | --- |
> | Pre-training Performance (Avg.) | 45.8 | 44.9 | 46.0 | 47.2 | 47.5 | 48.2 |
> | Post-training Performance (Avg.) | 53.2 | 50.0 | 49.1 | 58.4 | 59.9 | 65.4 |
>
> These results show that while ADO is a strong dynamic sampling baseline with negligible search overhead, **FastMix consistently achieves higher downstream performance** in both pre-training and post-training regimes, while still incurring only a small search cost (using a single proxy model). We will make these conceptual distinctions and empirical comparisons explicit in the revised manuscript, see the added content (blue colored) in the Appendix Section C.

---

> ### Author Response · Authors · 2025-11-25
> **Response to Weakness-2 by Reviewer PnEr**
>
> ### **Q2: Response to Weakness-2: The reviewer provides some reference works.**
>
> Thank you for pointing out these additional relevant works. We have incorporated all of them into the revised related work section and provide a more comprehensive contextualization of prior approaches, see the added content (blue colored) in the Appendix Section C.
>
> **Data Mixing Laws [2]** propose predictive *mixing functions* that estimate downstream performance as a function of mixture proportions. By fitting these functions using a small number of sampled mixtures, the method can *extrapolate* performance for unseen mixtures without fully training models on them. This approach focuses on *predictive modeling* of the mixture–performance relationship rather than differentiable optimization.
>
> **MFMS-GP [3]** formulates data mixture optimization as a *multi-fidelity, multi-scale Bayesian optimization* problem. Instead of relying on deterministic scaling-law curves, it adopts a probabilistic surrogate model that jointly captures uncertainty across mixtures, model sizes, and training steps. The search is viewed as *sequential decision-making* under uncertainty, enabling principled exploration of the mixture space.
>
> **ADMIRE-BayesOpt [4]** similarly treats mixture selection as a *black-box hyperparameter optimization* problem and applies multi-fidelity Bayesian optimization to reduce the cost of evaluating large models. ADMIRE explicitly studies *cross-scale transferability* of optimal mixtures and uses low-fidelity evaluations to accelerate the identification of promising mixture configurations for expensive large-scale training.
>
> Together with ADO [1], these works form an important line of research exploring *non-gradient*, *surrogate-based*, or *Bayesian* strategies for mixture optimization. In contrast, **FastMix takes a fully differentiable bilevel approach**, directly optimizing mixture weights through hyper-gradients of the validation loss using a single proxy model, which substantially reduces computational overhead while providing task-driven mixture updates.
>
> ---
>
>
> [1] Adaptive Data Optimization: Dynamic Sample Selection with Scaling Laws
>
> [2] Data Mixing Laws: Optimizing Data Mixtures by Predicting Language Modeling Performance
>
> [3] Data Mixture Optimization: A Multi-fidelity Multi-scale Bayesian Framework
>
> [4] ADMIRE-BayesOpt: Accelerated Data MIxture RE-weighting for Language Models with Bayesian Optimization

---

> ### Author Response · Authors · 2025-11-25
> **Response to Weakness-3 by Reviewer PnEr**
>
> ### **Response to Weakness-3: The reviewer wishes to see the Standard Deviation of the main results.**
>
> We appreciate the reviewer’s concern regarding uncertainty estimates and the robustness of our reported gains. In response, we have run all methods (except DoReMi and ODM, for which we use the official results from RegMix) with multiple random seeds and now report mean ± standard deviation for both the pre-training and post-training settings.
>
> **1. Pre-training: mean ± standard deviation**
>
> The table below summarizes the performance of FastMix and baselines under the pre-training setting, together with their standard deviations. Overall, FastMix:
>
> - Achieves the best **average performance** (48.2 ± 0.5) and **best average rank (1st)**.
> - Attains the **lowest variance** among mixture-optimization methods (RegMix, CLIMB, IDEAL), indicating that its gains are stable rather than seed-specific.
> - Achieves this while using only $\frac{1}{550}$ of RegMix’s search cost and$\frac{1}{55}$ of CLIMB’s.
>
> The results for DoReMi and ODM are copied from RegMix; since their performance is substantially worse, we did not consume additional compute to re-run them with multiple seeds.
>
>
> | **Benchmark** | **DoReMi** | **ODM** | **IDEAL** | **RegMix** | **CLIMB** | **FastMix (Ours)** |
> | --- | --- | --- | --- | --- | --- | --- |
> | Social IQA | 33.4 | 33.7 | 32.5$\pm4.5$ | 33.8$\pm0.7$ | **34.2$\pm1.3$** | 33.6$\pm0.9$ |
> | HellaSwag | 43.4 | 37.2 | 41.9$\pm9.1$ | 44.2$\pm5.2$ | 43.4$\pm3.7$ | **44.7$\pm1.1$** |
> | PiQA | 68.3 | 64.4 | 64.9$\pm5.4$ | 68.0$\pm4.4$ | 67.9$\pm3.9$ | **69.8$\pm1.2$** |
> | OpenBookQA | 30.3 | 30.0 | 29.6$\pm1.6$ | 30.3$\pm2.7$ | 29.3$\pm2.1$ | **31.5$\pm0.8$** |
> | Lambada | 32.1 | 29.6 | 35.1$\pm6.4$ | 34.2$\pm7.3$ | 35.9$\pm4.1$ | **36.3$\pm1.1$** |
> | SciQ | 81.6 | 79.8 | 75.3$\pm5.3$ | **82.8$\pm3.1$** | 82.4$\pm1.3$ | 82.2$\pm0.9$ |
> | ARC Easy | 50.6 | 47.9 | 50.1$\pm4.2$ | 51.7$\pm3.6$ | 51.1$\pm2.4$ | **52.5$\pm2.0$** |
> | ARC Challenge | 26.1 | 25.6 | 25.3$\pm2.6$ | 25.7$\pm5.6$ | 25.2$\pm6.1$ | **27.0$\pm3.7$** |
> | COPA | 68.5 | 68.2 | 65.1$\pm5.7$ | 70.2$\pm4.5$ | **70.7$\pm2.8$** | 70.5$\pm1.6$ |
> | RACE | 31.3 | 29.7 | 30.3$\pm5.2$ | 31.3$\pm2.3$ | 30.9$\pm3.5$ | **31.6$\pm0.9$** |
> | LogiQA | 26.4 | 25.6 | 26.9$\pm3.1$ | 25.8$\pm2.2$ | 27.7$\pm1.7$ | **28.1$\pm1.6$** |
> | QQP | 56.6 | 53.1 | 47.0$\pm6.0$ | 58.3$\pm5.9$ | **58.6$\pm1.9$** | 58.4$\pm1.1$ |
> | WinoGrande | 52.2 | 51.8 | 71.7$\pm4.2$ | 53.1$\pm1.5$ | 54.7$\pm1.1$ | **55.7$\pm0.7$** |
> | MultiRC | 53.8 | 53.3 | 49.4$\pm2.1$ | 51.7$\pm4.0$ | 53.2$\pm2.9$ | **53.4$\pm3.4$** |
> | **Average Performance (↑)** | 46.8 | 44.9 | 46.0$\pm4.4$ | 47.2$\pm3.8$ | 47.5$\pm2.4$ | **48.2$\pm0.5$** |
> | **Average Rank (↓)** | 4 | 6 | 5 | 3 | 2 | **1** |
> | **Time-cost (GPU Hours) in Searching (↓)** | 7.4 | $\approx$0 | $\approx$0 | 720.5 | 71.9 | **1.3** |
>
> We also note that FastMix achieves the best score on 10 out of 14 benchmarks and is competitive (within one standard deviation) on the remaining ones, which indicates that the average improvement of +0.7 over CLIMB is both consistent and larger than the residual run-to-run variability.
>
> **2. Post-training: mean ± standard deviation**
>
> For the post-training setting, we again evaluate all methods with multiple seeds and report mean ± standard deviation. FastMix not only delivers the highest average performance but also the smallest variance, while requiring only **2.2 GPU hours**, i.e., about $\frac{1}{115}$ of the search compute used by RegMix and CLIMB.
>
>
> | **Benchmark** | **IDEAL** | **RegMix** | **CLIMB** | **FastMix (Ours)** |
> | --- | --- | --- | --- | --- |
> | MATH | 86.2$\pm4.3$ | 89.4$\pm6.8$ | 91.1$\pm4.7$ | **93.1$\pm3.9$** |
> | AIME-24 | 30.0$\pm2.1$ | 36.6$\pm4.2$ | 43.3$\pm3.8$ | **53.3$\pm2.4$** |
> | LiveCodeBench | 27.9$\pm6.9$ | 58.3$\pm3.9$ | 60.2$\pm5.0$ | **62.5$\pm3.2$** |
> | GPQA-Diamond | 52.3$\pm5.4$ | 49.4$\pm4.1$ | 45.0$\pm3.1$ | **52.8$\pm2.7$** |
> | **Average Performance (↑)** | 49.1$\pm4.7$ | 58.4$\pm4.8$ | 59.9$\pm4.2$ | **65.4$\pm3.0$** |
> | **Time-cost (GPU Hours) in Searching (↓)** | ≈0 | 115.9 | 117.4 | 2.2 |
>
>
> These additional results directly address the reviewer’s concern: FastMix’s gains are not due to a favorable seed but are statistically robust across multiple runs, while also being dramatically more compute-efficient than prior mixture-optimization methods.

---

> ### Author Response · Authors · 2025-11-25
> **Response to Questions by Reviewer PnEr**
>
> ### **Response to Question-1: Clarification on the “search target” and gradient computation**
>
> Thank you for this insightful question. To clarify:
>
> **(1) What is the “search target”?**
>
> Equation 7 defines the **full unified search objective** used in all our main experiments. It consists of three components:
>
> - the **validation loss**,
> - the **training loss**, scaled by β, and
> - the **entropy regularization**, scaled by λ.
>
> This combined objective is what the algorithm optimizes during mixture search.
>
> **(2) Relationship between Eq. 6 and Eq. 7**
>
> Equation 6 is a **special case** of Equation 7 obtained by setting β = 0 and λ = 0.
>
> We first presented Eq. 6 to introduce the core idea in its simplest form (pure validation-loss optimization), and then introduced Eq. 7 to describe the enhanced version used in practice.
>
> **(3) Which gradient was actually computed?**
>
> In all reported experiments, the hyper-gradient was computed using the **entire objective in Eq. 7**, i.e., including the validation term, the β-weighted training-loss term, and the λ-weighted entropy term.
>
> To avoid confusion, we have revised the manuscript to make the relationship between Eq. 6 and Eq. 7 explicit, and to state clearly that Eq. 7 is the true search objective throughout the experiments.
>
> ---
>
> ### **Response to Question-2 and 3: Details on random initialization in Fig. 2(c)**
>
> Thank you for the question. In Fig. 2(c), *“random initialization”* refers specifically to **different random initializations of the mixture weights**. Across 11 independent runs (E0–E10), we observe a mean performance of **48.34** with a small standard deviation of **0.48**, indicating that FastMix is highly robust to initialization—consistent with the observations reported in Section 4.3.3.
>
> Our mixture initialization procedure is as follows:
>
> 1. **Independent sampling:** Each mixture coefficient is independently sampled from a Uniform[0,1] distribution.
> 2. **Projection onto the simplex:** After sampling, we normalize the entire vector by dividing each coefficient by the sum of all coefficients. This ensures that the initialized mixture lies on the probability simplex (i.e., non-negative and sums to 1), yielding a valid distribution over datasets.
>
> This procedure produces a diverse set of valid initial mixtures while ensuring feasibility. The small variation across runs demonstrates that FastMix’s optimization dynamics are stable and not sensitive to the randomness in mixture initialization.
>
>
> ---
>
> ### **Response to Question-4. Convergence behavior and sensitivity to initial mixtures**
>
> Thank you for raising these points. We are happy to clarify.
>
> **(1) Does the mixture optimization converge to a stable distribution?**
>
> Yes. In practice, FastMix converges to a stable mixture well before training ends. Across all experiments, the mixture weights typically stabilize within the first **10–40%** of the optimization steps. After this point, updates become very small, and the mixture proportions remain effectively unchanged.
>
> **(2) How sensitive is FastMix to the initial mixture?**
>
> FastMix is highly robust to initialization. As shown in Fig. 2(c), 11 independent runs (E0–E10) with different random initial mixtures converge to almost identical performance, with a mean of **48.34** and a standard deviation of only **0.48**. This consistency indicates that:
>
> - The optimization landscape has a **dominant basin of attraction**, and
> - different initial mixtures reliably converge toward the same final solution.
>
> **(3) Visualization of mixture trajectories**
>
> While the main paper focuses on final performance due to space constraints, we do provide a visualization of the **final learned mixtures** in Appendix A (Figure 3). These distributions are smooth, interpretable, and aligned with the performance improvements reported in the main text.
>
> We also have internal logs showing the **full trajectory curves** for mixture updates. These trajectories exhibit rapid convergence early in training and remain stable thereafter. We have updated the appendix to include additional trajectory plots to make this behavior explicit; see the added content (blue colored) in Appendix Section A.

---

### Official Review · Reviewer_Tw57 · 2025-11-02

**Soundness:** 3
**Presentation:** 2
**Contribution:** 3
**Rating:** 6
**Confidence:** 2

**Summary:**

The paper tackles the problem of data mixture optimization for LLM pre-training and SFT. Existing strong methods such as RegMix and CLIMB find good mixtures but do so by training many proxy models, which makes them expensive and slow to iterate on.  The paper proposes FASTMIX, which makes mixture search differentiable by (i) reformulating mixture selection as a weighted bilevel optimization problem and (ii) showing that sampling from a mixture is equivalent in expectation to doing uniform source sampling but applying per-source loss weights. This reparameterization allows the authors to update both model weights and mixture weights in the same training loop with a lightweight hyper-gradient.

The paper claims:
	1.	On pre-training over Pile-style subsets, FASTMIX achieves the best average score (48.2) at ~1.3 GPU-hours of search, which the authors say is 55× faster than CLIMB and 550× faster than RegMix while still slightly better in accuracy.
	2.	On post-training/SFT (math-anchored, tested on coding & STEM QA), FASTMIX reaches 65.4, about +5.5 over the next best method, in 2.2 GPU-hours.

So the core contribution is a practical, differentiable, single-proxy recipe for data-mixture search that aims to match proxy-based methods in quality but at 1–2 GPU-hours instead of 100+.

**Strengths:**

1. The key equivalence of “sampling by α” ≡ “uniform source sampling + per-source loss weights α” is written down cleanly and plugged straight into a bilevel objective. This makes the method implementable in an existing LLM training codebase with only per-batch source tags and a vector of α.
2. Compute saving is compelling. The claim “1.3 GPU-hours vs 55–550× more for baselines” is helpful, because current RegMix-style methods really do train large banks of proxies. The authors also highlight that they train only one proxy model.

**Weaknesses:**

No uncertainty / significance reporting. Gains like “+0.7 to +1.0” in pre-training are plausible but small. Without multiple seeds or CIs, especially on the final model, it’s hard to tell whether the improvement is robust or just a friendly seed. This matters because RegMix/CLIMB papers do report variability.

**Questions:**

Can the method enforce per-source lower/upper bounds (e.g. “at least 10% safety-filtered”, “at most 5% web-crawl”)? If yes, how does that change the hyper-gradient update?

---

> ### Author Response · Authors · 2025-11-25
> **Response to Weakness by Reviewer Tw57**
>
> Thank you for your valuable comments and the positive evaluation of our work! We are more than happy to address any questions you may have. If you have any further concerns or suggestions, please feel free to let us know!
>
> ---
>
>
> ### **Q1. Response to Weakness: The reviewer wishes to see the Standard Deviation of the main results.**
>
> We appreciate the reviewer’s concern regarding uncertainty estimates and the robustness of our reported gains. In response, we have run all methods (except DoReMi and ODM, for which we use the official results from RegMix) with multiple random seeds and now report mean ± standard deviation for both the pre-training and post-training settings.
>
> **1. Pre-training: mean ± standard deviation**
>
> The table below summarizes the performance of FastMix and baselines under the pre-training setting, together with their standard deviations. Overall, FastMix:
>
> - Achieves the best **average performance** (48.2 ± 0.5) and **best average rank (1st)**.
> - Attains the **lowest variance** among mixture-optimization methods (RegMix, CLIMB, IDEAL), indicating that its gains are stable rather than seed-specific.
> - Achieves this while using only $\frac{1}{550}$ of RegMix’s search cost and$\frac{1}{55}$ of CLIMB’s.
>
> The results for DoReMi and ODM are copied from RegMix; since their performance is substantially worse, we did not consume additional compute to re-run them with multiple seeds.
>
>
> | **Benchmark** | **DoReMi** | **ODM** | **IDEAL** | **RegMix** | **CLIMB** | **FastMix (Ours)** |
> | --- | --- | --- | --- | --- | --- | --- |
> | Social IQA | 33.4 | 33.7 | 32.5$\pm4.5$ | 33.8$\pm0.7$ | **34.2$\pm1.3$** | 33.6$\pm0.9$ |
> | HellaSwag | 43.4 | 37.2 | 41.9$\pm9.1$ | 44.2$\pm5.2$ | 43.4$\pm3.7$ | **44.7$\pm1.1$** |
> | PiQA | 68.3 | 64.4 | 64.9$\pm5.4$ | 68.0$\pm4.4$ | 67.9$\pm3.9$ | **69.8$\pm1.2$** |
> | OpenBookQA | 30.3 | 30.0 | 29.6$\pm1.6$ | 30.3$\pm2.7$ | 29.3$\pm2.1$ | **31.5$\pm0.8$** |
> | Lambada | 32.1 | 29.6 | 35.1$\pm6.4$ | 34.2$\pm7.3$ | 35.9$\pm4.1$ | **36.3$\pm1.1$** |
> | SciQ | 81.6 | 79.8 | 75.3$\pm5.3$ | **82.8$\pm3.1$** | 82.4$\pm1.3$ | 82.2$\pm0.9$ |
> | ARC Easy | 50.6 | 47.9 | 50.1$\pm4.2$ | 51.7$\pm3.6$ | 51.1$\pm2.4$ | **52.5$\pm2.0$** |
> | ARC Challenge | 26.1 | 25.6 | 25.3$\pm2.6$ | 25.7$\pm5.6$ | 25.2$\pm6.1$ | **27.0$\pm3.7$** |
> | COPA | 68.5 | 68.2 | 65.1$\pm5.7$ | 70.2$\pm4.5$ | **70.7$\pm2.8$** | 70.5$\pm1.6$ |
> | RACE | 31.3 | 29.7 | 30.3$\pm5.2$ | 31.3$\pm2.3$ | 30.9$\pm3.5$ | **31.6$\pm0.9$** |
> | LogiQA | 26.4 | 25.6 | 26.9$\pm3.1$ | 25.8$\pm2.2$ | 27.7$\pm1.7$ | **28.1$\pm1.6$** |
> | QQP | 56.6 | 53.1 | 47.0$\pm6.0$ | 58.3$\pm5.9$ | **58.6$\pm1.9$** | 58.4$\pm1.1$ |
> | WinoGrande | 52.2 | 51.8 | 71.7$\pm4.2$ | 53.1$\pm1.5$ | 54.7$\pm1.1$ | **55.7$\pm0.7$** |
> | MultiRC | 53.8 | 53.3 | 49.4$\pm2.1$ | 51.7$\pm4.0$ | 53.2$\pm2.9$ | **53.4$\pm3.4$** |
> | **Average Performance (↑)** | 46.8 | 44.9 | 46.0$\pm4.4$ | 47.2$\pm3.8$ | 47.5$\pm2.4$ | **48.2$\pm0.5$** |
> | **Average Rank (↓)** | 4 | 6 | 5 | 3 | 2 | **1** |
> | **Time-cost (GPU Hours) in Searching (↓)** | 7.4 | $\approx$0 | $\approx$0 | 720.5 | 71.9 | **1.3** |
>
> We also note that FastMix achieves the best score on 10 out of 14 benchmarks and is competitive (within one standard deviation) on the remaining ones, which indicates that the average improvement of +0.7 over CLIMB is both consistent and larger than the residual run-to-run variability.
>
> **2. Post-training: mean ± standard deviation**
>
> For the post-training setting, we again evaluate all methods with multiple seeds and report mean ± standard deviation. FastMix not only delivers the highest average performance but also the smallest variance, while requiring only **2.2 GPU hours**, i.e., about $\frac{1}{115}$ of the search compute used by RegMix and CLIMB.
>
>
> | **Benchmark** | **IDEAL** | **RegMix** | **CLIMB** | **FastMix (Ours)** |
> | --- | --- | --- | --- | --- |
> | MATH | 86.2$\pm4.3$ | 89.4$\pm6.8$ | 91.1$\pm4.7$ | **93.1$\pm3.9$** |
> | AIME-24 | 30.0$\pm2.1$ | 36.6$\pm4.2$ | 43.3$\pm3.8$ | **53.3$\pm2.4$** |
> | LiveCodeBench | 27.9$\pm6.9$ | 58.3$\pm3.9$ | 60.2$\pm5.0$ | **62.5$\pm3.2$** |
> | GPQA-Diamond | 52.3$\pm5.4$ | 49.4$\pm4.1$ | 45.0$\pm3.1$ | **52.8$\pm2.7$** |
> | **Average Performance (↑)** | 49.1$\pm4.7$ | 58.4$\pm4.8$ | 59.9$\pm4.2$ | **65.4$\pm3.0$** |
> | **Time-cost (GPU Hours) in Searching (↓)** | ≈0 | 115.9 | 117.4 | 2.2 |
>
> ---
>
> These additional results directly address the reviewer’s concern: FastMix’s gains are not due to a favorable seed but are statistically robust across multiple runs, while also being dramatically more compute-efficient than prior mixture-optimization methods.

---

> ### Author Response · Authors · 2025-11-25
> **Response to Question by Reviewer Tw57**
>
> ### **Q2. Response to Question: Can the method enforce per-source lower/upper bounds? If yes, how does that change the hyper-gradient update?**
>
> This is an excellent question, and we appreciate the opportunity to clarify. Our main experiments do not impose per-source lower or upper bounds, allowing the optimizer to freely discover the most effective mixture. However, **FastMix is fully compatible with per-source constraints**, and the modification is straightforward.
>
> Incorporating bounds simply requires adding a **projection step** after each hyper-gradient update. Concretely, after updating the mixture coefficients, we:
>
> 1. **Clip** any source whose value violates its lower or upper bound to the nearest feasible value;
> 2. **Renormalize** the remaining free coefficients so that the mixture still sums to 1.
>
> This projection procedure does **not** change the form of the hyper-gradient—it leaves the gradient computation untouched—but it **ensures feasibility at every iteration** by constraining α\alphaα after the update. In other words, the hyper-gradient stays exactly the same; we only modify the optimization trajectory through projection.
>
> We also tested this setting on both RegMix and FastMix. RegMix first trains a regression-based surrogate model that maps mixtures to performance and only then performs a constrained search on top of this surrogate. As a result, **RegMix can only enforce constraints at the final search stage**, not throughout training.
>
> In contrast, FastMix applies constraints **continuously throughout optimization**, which keeps the trajectory inside the feasible region and avoids the instability or suboptimality that may arise when constraints are imposed only at the end.
>
>
> ---
>
> **Empirical comparison under per-source bounds**
>
> In our pre-training setting, we applied the following constraints:
>
> - **Pile-cc ≤ 0.84** since we found this source is important (also as in RegMix paper)
> - **All other sources ≥ 0.01**
>
>
>
> The resulting performance is shown below:
>
> | Method | RegMix | Ours | Ours (w/o constraint) |
> | --- | --- | --- | --- |
> | Performance | 44.3 | 47.1 | 48.2 |
>
> FastMix remains substantially better under constraints. We also note that the constrained performance (47.1) is slightly below the unconstrained result reported in our main paper (48.2), which is expected: the constraints reduce the feasible mixture space. This further highlights that **FastMix effectively leverages the full data distribution when allowed, but remains robust and performant even under strict source-level bounds**.

---

### Author Response · Authors · 2025-11-27
**Summary of Responses**

We sincerely thank the reviewers for their time, constructive feedback, and positive evaluation of our work. Below is a summary of the key concerns addressed and the new experimental evidence provided in the rebuttal.

---



Reviewer Index, Question/Weakness Index | Main Questions | Response
---|---|---
Tw57 W1, PnEr W3 | Statistical Robustness and Variance Analysis | We ran all major experiments (pre-training and post-training) with multiple random seeds and now report the mean $\pm$ standard deviation ($\mu \pm$ SD) for all methods. FastMix not only achieves the best average performance but also exhibits the lowest variance among all mixture-optimization methods.
PnEr W1\&W2 | Some new related works | We have discussed the differences between ours and theirs. We have also add this in the revision.
Kg4B W4 | Scaling-up Baselines Performance | We dramatically scaled up the search complexity for RegMix and CLIMB by increasing the proxy model number from 64 to 512 in the post-training setting (an 8x increase in computational cost, reaching $940+$ GPU-Hours). However, the baselines still lag significantly behind FastMix's performance, which requires only 2.2 GPU-Hours for the search.
Kg4B W2\&W7\&Q1\&Q6 | The reviewer wishes to see how to handle non-smooth and non-differentiable objectives (e.g., Accuracy). |  We propose and test a Hybrid Optimization Strategy: use a fast, differentiable proxy (SFT Loss) in the early phase, then switch to the non-differentiable target (Accuracy) with a finite-difference estimator in the later phase. The Hybrid strategy achieved the best performance ($\mathbf{66.0}$), proving that FastMix can effectively optimize non-smooth metrics without relying solely on computationally expensive finite-difference methods.
Kg4B W3, Q2 | Cross-Model-Size Generalization | The optimal mixture found by FastMix shows excellent cross-size generalization. The performance on the 32B model remains almost identical whether the search was performed on a 1M, 0.5B, or 1B proxy model, ensuring that our highly efficient search method is reliable for production-scale LLMs.
Kg4B W5, Q8 | Cross-Tasks Generalization and overfitting concerns | We demonstrated that in the Post-training setting, even when strictly using Math reasoning ability as the sole search objective, the mixture strategy identified by FastMix enables the model to achieve leading performance in other crucial domains like Code and STEM, surpassing all baselines, including RegMix and CLIMB. Furthermore, we rigorously tested the impact of regularization by conducting two comparative experiments: (1) removing the regularization term from FastMix, and (2) integrating our regularization objective into all baselines (such as RegMix). We consistently found that FastMix's performance advantage remains uniformly stable and consistent across all these regularization schemes.
Kg4B W1, Q3 | A dynamic version of our FastMix | We explored a simple dynamic extension by updating the mixture every $K$ training steps. Key Conclusions: **Overly Dynamic Strategies Fail:** The most dynamic strategy ($K=1$) proved detrimental, yielding the poorest performance (41.5), indicating frequent, real-time changes can be highly destabilizing. **Optimal Balance:** Optimal performance ($\mathbf{48.9}$) was achieved at an intermediate update frequency ($K = \text{max-iter} / 50$), only a marginal gain over the totally-fixed policy ($\mathbf{48.2}$). **Implication:** The most dynamic approach is not the best. An optimal balance exists between fully static and fully dynamic regimes. We commit to investigating more sophisticated dynamic strategies in future work.
Kg4B W6, Q4 | Multi-modal Experiments | We addressed the concern regarding FastMix's scalability beyond text by conducting experiments in a multimodal image–text pre-training setting using the LAION-400M dataset. We optimized mixture weights over 512 visual clusters derived from image features (DINO) of LAION-400M. The search used a small ViT-Tiny/BERT proxy model (50M parameters) with CIFAR-100 zero-shot accuracy as the objective. Key Conclusions: **Generalization Test:** Evaluation was performed on a much larger **ViT-Large model** across four different metrics (ImageNet zero-shot/linear-probe, Flickr30K retrieval). **Leading Performance:** FastMix consistently **outperformed all baselines** (RegMix and CLIMB) in this multimodal setting, achieving the highest average performance ($\mathbf{59.3}$). **Robustness:** Even **without regularization**, FastMix showed superior results ($\mathbf{59.0}$ Avg.), demonstrating that its core differentiable formulation generalizes effectively to complex multimodal data mixtures and scales reliably from small proxy models to large vision–language models.
Kg4B  Q5, Q7|Regularization terms' ablation, and Baselines with our regularizations.|Even without regularization, FastMix still outperforms all others (including other baselines with our proposed regularization terms)

---

### Author Response · Authors · 2025-12-01
**Summary of all rebuttals and states.**

Dear AC:

We sincerely thank you and all the reviewers for your efforts! Below is a summary of our rebuttal for your review.



Reviewer|Strength|Questions|Initial-rate|Our reply
---|---|---|---|---
Tw57|**1**. Formulation is novel.   **2**. FastMix is significantly efficient|Wish to see 🔥**A**.the standard deviation   🔥**B**.performance under per-source bounded constraints|MA:6|**A**.✅ We have provided the standard deviation information.   **B**.✅ We have demonstrated that FastMix easily achieves leading performance even with these conditions.
PnEr|**1**. Strong results.   **2**. Novel Formulation. Clear and insightful intuition.   **3**. Sufficient ablation studies. |🔥**A.** Some additional related works.   🔥**B.** Wish to see the standard deviation   🔥**C.** Further explain the search target, details in the random initialization.   🔥**D.** How does FastMix converge?|MA:6|**A**.✅ We added the new related-work discussion in the revision (Appendix Sec.C).   **B**.✅ We have provided the standard deviation.   **C**.✅ We made further explanations about the questions.   **D**.✅ We provide an analysis and a plot figure of the searching dynamics in Appendix Sec.A.
Kg4B| **1**. Interesting formulation. **2.** Significant performance gains and time savings. **3**. Practicality and Simplicity. **4**. The use of a single proxy model against hundreds is appreciable. |🔥**A**. Wish to see Baselines‘ Performance by Scaling-up proxy model number to 512, see Weakness-4.  🔥**B**. Wish to see how to handle non-smooth and non-differentiable objectives (e.g., Accuracy), see Weakness-2/7 and Question-1/6.  🔥**C**. Wish to see Cross-Model-Size Generalization Performance, See W3 and Q2.  🔥**D**. Wish to see Cross-Tasks Generalization (**already in the original paper!**), see W5, and Q8  🔥**E**. Wish to see whether FastMix could be adapted as a dynamic data mixing method, see W1 and Q3.  🔥**F**. Wish to see experiments on large-scale multi-modal settings, see W6 and Q4.  🔥**G**. Wish to see the regularization terms' ablation, and how baseline methods perform when with our regularization terms, see Q5 and Q7. | MR: 4| **A**.✅ Provided! And FastMix is still leading!  **B**.✅ We provided a fast and effective hybrid method to handle the non-smooth and non-differentiable target (searching with a differentiable target first, then using the non-smooth target with the finite difference gradient estimator).   **C**.✅ We provide the cross-size generalization test from 1M, 0.5B, or 1B proxy models to large models as large as 32B. The final results are very robust and consistent.  **D**.✅ Already in the original paper, and we restate it.  **E**.✅ We explored a simple dynamic extension by updating the mixture every $K$ training steps. Key Conclusions: **Overly Dynamic Strategies Fail:** The most dynamic strategy ($K=1$) proved detrimental, yielding the poorest performance (41.5), indicating frequent, real-time changes can be highly destabilizing. **Optimal Balance:** Optimal performance ($\mathbf{48.9}$) was achieved at an intermediate update frequency ($K = \text{max-iter} / 50$), only a marginal gain over the totally-fixed policy ($\mathbf{48.2}$). **Implication:** The most dynamic approach is not the best. An optimal balance exists between fully static and fully dynamic regimes. We commit to investigating more sophisticated dynamic strategies in future work.  **F**.✅ We provided the results on large-scale vision-language pretraining (LAION-400M)! FastMix still outperforms all others.    **G**.✅ Even without the regularization term, FastMix still outperforms all others (including other baselines with our proposed regularization terms)

---

### Meta-Review · Area_Chair_hSsd · 2026-01-12

**Summary:**

The paper proposes a differentiable framework for large-scale model training. It determines how to efficiently determine the optimal proportions of multiple data sources to maximize performance on a target validation objective: both for pre-training and post-training (SFT) stages. The cast the problem as bilevel optimization problem, which significantly reduces the resource consumption. The reviewers were generally positive. Especially, they showed that across pre- and post-training, their method outperforms several baselines by a signifcant amount. A reduction of 2.2 GPU hours from 115 GPU hours  as compared to the baselines is very impressive.
Reviewer Kg4B wrote very detailed review and asked several questions, which are adequately addressed by the authors.

I recommend acceptance.

**Reviewer Concerns:**

Authors have adequately addressed reviewer concerns.

**Reviewer Scores:**

I believe that given authors concerns, Reviewer Kg4B would have increased their score.

---

### Decision · Program_Chairs · 2026-01-26

Accept (Poster)